



# A statistical analysis of zircon age distributions in volcanic, porphyry and plutonic rocks

Chetan Nathwani, Dawid Szymanowski, Lorenzo Tavazzani, Sava Markovic, Adrianna L. Virmond, and Cyril Chelle-Michou

Institute of Geochemistry and Petrology, ETH Zürich, Zürich, Switzerland

**Correspondence:** Chetan Nathwani (chetan.nathwani@eaps.ethz.ch)

**Abstract.** The distribution of zircon crystallisation ages in igneous rocks has been proposed to provide insights into the dynamics of underlying magma reservoirs. However, the ability to interpret magmatic processes from an age distribution is challenged by a complex interplay of factors such as sampling biases, analytical uncertainties and incorporation of extraneous zircon grains. Here, we used a compilation of magmatic zircon U-Pb ages measured by chemical abrasion isotope dilution thermal ionisation mass spectrometry (CA-ID-TIMS) to quantify the differences that exist among zircon U-Pb age distributions
from different magmatic systems. The compiled dataset was rigorously filtered through a number of processing steps to isolate age distributions least impacted by sampling biases and analytical factors. We also filter the database using a new algorithm to systematically identify and remove old outliers from age distributions. We adopt the Wasserstein distance as a dissimilarity metric to quantify the difference between the shapes of age distributions. Principal component analysis of a dissimilarity matrix
of pairwise Wasserstein distances of age distributions reveals a difference between zircon age distributions found in plutonic, porphyry and volcanic rocks. Volcanic and porphyry zircon populations exhibit a skew towards younger ages in their distributions, whereas plutonic age distributions skew towards older ages. Using a bootstrap sampling approach to generate synthetic age distributions, we show that this difference can be predominantly ascribed to truncation of zircon crystallisation during volcanic eruptions and porphyry dyke emplacement, which leads to a younger skew. We also find that higher magmatic flux
can contribute to the younger skew of volcanic and porphyry zircon age distributions, though we emphasise that no difference in flux is required given the strong effect of truncation on zircon age distributions. Given the multitude of factors that influence zircon age distributions, we urge caution when quantifying the thermal evolution of crustal magma bodies using zircon age distributions integrated with numerical models.

## 1 Introduction

U–Th–Pb geochronology of zircon in igneous rocks provides key information about the age, longevity, and emplacement rates of magma reservoirs. Historically, the achievable age resolution limited these insights to a singular 'age' of a magmatic system, but as analytical precision and accuracy have improved, it has become possible to resolve extended records of zircon crystallisation within a single igneous rock sample. These zircon crystals can predate the eruption or final solidification of a magma body by as much as a million years (Brown and Fletcher, 1999; Wotzlaw et al., 2013; Barboni et al., 2015; Samperton





et al., 2015; Szymanowski et al., 2019). The observed zircon populations may result from cooling of the magma volume in which they are found (i.e. they are autocrystic, Wotzlaw et al., 2013; Samperton et al., 2015) or they could reflect the juxtaposition of zircon populations that derive from multiple levels within the crustal column (e.g., Schoene et al., 2012; Farina et al., 2018). While still relatively under-explored, distributions of zircon ages are promising recorders of processes that are critical to the ultimate fate of the host magmatic system (e.g. a volcanic eruption or economic mineralisation). In

several studies, zircon age distributions have been shown to match those produced from zircon solubility models and thus have been interpreted as the product of monotonous cooling of the magmatic system (Samperton et al., 2017; Keller et al., 2017). Others have documented age distributions which differ from these zircon solubility models and have attributed this to competition between cooling and recharge of the magma reservoir which shifts the peak in the distribution to younger ages (Keller et al., 2018; Tavazzani et al., 2023a). Coupling of zircon age distributions with outputs from numerical models has been

used to quantify greater magmatic fluxes related to magma reservoirs forming super-eruptions compared to plutonic complexes (Caricchi et al., 2014, 2016; Weber et al., 2020; Schmitt et al., 2023).

Isolating the effect of cooling and recharge processes on zircon age distributions can be challenging since a number of analytical and geological factors may play a role (Klein and Eddy, 2023). Firstly, a key requirement of this comparison is the ability to confidently resolve differences in crystallisation age within a single rock sample, i.e., to ensure that the observed

distribution is controlled by geological dispersion rather than analytical uncertainty. This ability, best described by the observed duration of zircon crystallisation with respect to the size of average analytical uncertainties of a dataset ($\Delta t/\sigma$), varies with the employed analytical technique, the time range of zircon crystallisation and absolute age. Datasets also contain variable numbers of zircon dates per sample and the ability to accurately capture the underlying age distribution increases with the number of zircons analysed (Tavazzani et al., 2023a). Interpreting age spectra is further challenged by truncations in zircon

crystallisation whereby dyke emplacement or volcanic eruption terminates zircon crystallisation at intermediate crystallinity, whereas in plutonic systems zircon crystallisation likely continues until the solidus (Samperton et al., 2017; Ratschbacher et al., 2018). Comparing age distributions is also grounded on the assumption that the analysed zircons are native to the investigated magma and are not a cargo of zircons crystallised in multiple, discrete systems incorporated upon transport (typically termed "antecrysts"; Miller et al., 2007).

As the geochronology community presents a growing number of zircon age distributions from different magmatic systems with sufficiently long duration to analytically resolve an age distribution, constraining the controls on their distributions is becoming increasingly relevant. The number of available datasets has now become sufficient to perform systematic analyses of published data to identify patterns that can be meaningfully attributed to geological processes. Such a comparison requires a robust statistical approach which is capable of comparing distributions with varying dataset size and analytical uncertainty

without making assumptions about the shape of the distribution. Use of dissimilarity metrics, such as the Kolmogorov-Smirnov and Wasserstein distances, are becoming increasingly popular to compare age distributions, with successful applications to tracing sediment provenance in multi-sample datasets (Vermeesch, 2013; Lipp and Vermeesch, 2023). These approaches are objective and they can be applied pairwise to datasets with infinite numbers of distributions and can be visualised using dimensionality reduction techniques (Vermeesch, 2013).





**Table 1.** Symbols used in this study and their definitions

| Symbol | Definition |
| --- | --- |
| $\Delta t$ | Absolute duration of zircon crystallisation |
| $\Delta t_{rel}$ | Time between two zircon crystallisation distributions |
| $t_i$ | Time of crystallisation of $i$th zircon |
| $t_{rel}$ | Relative time of zircon crystallisation |
| $t_{sat}$ | Time of initial zircon saturation |
| $t_{end}$ | Time of termination of zircon crystallisation |
| $\sigma$ | Uncertainty |
| ECDF | Empirical cumulative distribution function |
| KDE | Kernel density estimation |
| $\hat{F}_n(t)$ | Interpolated ECDF of $t$ |
| $f_{xtal}$ | Zircon age probability density function |
| $W_p$ | The $p$th Wasserstein distance |
| $n_{zircon}$ | Number of zircon crystals |
| $n_{inh}$ | Number of inherited zircon crystals |
| $\delta$ | Delta function |
| $d$ | Dissimilarity matrix |
| $w_i$ | Weighting of $i$ |
| $\nabla \hat{F}_n(t)$ | Gradient of interpolated ECDF |
| $\nabla_M$ | Gradient below which an ECDF is marked as discontinuous |
| $t_{flat}$ | Relative time of ECDF below |
| $M^{-1}$ | Inverse function of M |

In this study, we compare zircon age spectra from 70 igneous units using the Wasserstein dissimilarity metric to constrain whether differences in age spectra may reflect variable dynamics of magmatic systems. We identify key differences between age distributions in plutonic, volcanic and porphyry lithologies and use a bootstrap modelling approach to explore the key factors controlling the variability of zircon age distributions in magmatic systems.

## 2   Methodology

### 2.1   Zircon U-Th-Pb age spectra and their visualisation

The high precision allowed by state-of-the-art U-Th-Pb dating techniques is showcased in rank-order plots of zircon dates from a single sample (Fig. 1A). In this scenario, zircon dates exhibit dispersion between the onset of zircon crystallisation (i.e. initial zircon saturation, $t_{sat}$) and the end of zircon crystallisation ($t_{end}$), which represents an eruptive event or the final solidification of





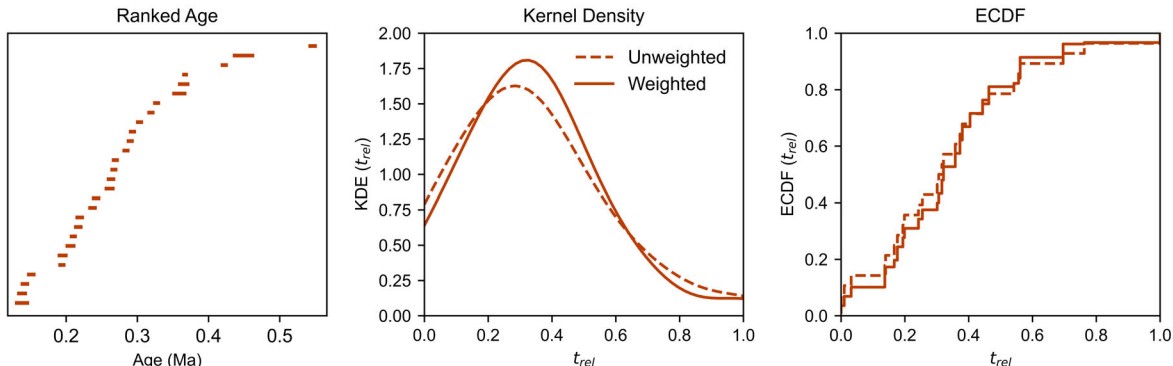

**Figure 1.** A comparison of three visualisation approaches employed in geochronology using an example zircon U-Pb ID-TIMS dataset from the Youngest Toba Tuff (Szymanowski et al., 2023). (A) Ranked age plot (youngest to oldest) where the horizontal extent of the bars indicates the $2\sigma$ uncertainty on each U-Pb date. (B) Kernel density estimate (KDE) using both unweighted (dashed) and weighted (solid) estimates and, (C) weighted (dashed) and unweighted (solid) empirical cumulative distribution frequency (ECDF) curves, both scaled between onset and end of zircon crystallisation ($t_{rel}$).

a magma batch. These are often treated as a scaled relative zircon crystallisation distribution, $f_{xtal}(t_{rel})$, where $t_{rel}$ is the relative

time scaled between $t_{sat}$ and $t_{end}$ (Keller et al., 2018):

$$t_{rel} = \frac{t - t_{end}}{t_{sat} - t_{end}} \tag{1}$$

The $f_{xtal}(t_{rel})$ of a sample can be visualised as a kernel density estimate (KDE; Fig. 1B) which is where a series of kernels (typically Gaussian) of a fixed width (the "bandwidith") are stacked along the distribution (Vermeesch, 2012). The main advantage of KDE plots is their ability to represent the distribution of zircon dates in an intuitive manner, moreover, they

can be weighted by the variable analytical uncertainties associated to each zircon date. Another method for visualising such age distributions is an empirical cumulative distribution function (ECDF; Fig. 1C), which is a step function that increases by $1/n$ (or by an interval inverse to the analytical uncertainty) at each of the $n$ dates. The ECDF, though less intuitive, presents several advantages. The first is that outliers (e.g. antecrysts) can be more easily identified because it is a step function that plots each discrete date unlike a KDE plot. The second is that it can be more intuitively related to dissimilarity metrics (e.g.

the Kolmogorov–Smirnov test and the Wasserstein distance) which are related to the distances between two ECDFs. We thus henceforth prefer to visualise age distributions as ECDFs.

## 2.2  Filtering of outliers in age distributions

The biggest obstacle to identifying a truly "magmatic" age distribution is the presence of entire zircons or zircon domains (e.g. crystal cores) which are foreign to the youngest magmatic event, introduced into the magma via assimilation/mixing during



emplacement or transport. Zircon crystals that formed within the youngest magmatic event are typically defined as "autocrysts",
and those that crystallise in an earlier magmatic event related to older pulses of the composite, longer-term magmatic system are
often termed "antecrysts" (Miller et al., 2007). "Xenocrysts" are those incorporated from host rocks and are typically millions
of years older, which makes them easier to exclude than antecrysts.

When analysing datasets of concordant U-Pb dates, identifying antecrysts in an age distribution is subjective. Many recent
interpretations of zircon age–composition datasets acknowledge that zircon populations found in individual igneous rocks com-
monly represent crystallisation in complex magmatic plumbing systems (e.g. Szymanowski et al., 2019; Pamukçu et al., 2022),
thereby making the autocryst–antecryst divide ambiguous, and possibly detrimental to the understanding of the underlying
system. Some authors may decide to exclude older tails of age distributions from their interpretation, but the criteria to do so
are variable and often not clearly outlined. In some studies this can be based on different trace element compositions of older
zircon, inferring they were derived from an unrelated magmatic event (e.g., Gagnevin et al., 2010; Siégel et al., 2018; Tavazzani
et al., 2023a). Other studies identify "breaks" in the age distribution which indicate that the older zircon crystals were derived
from a different source (e.g., Samperton et al., 2015).

In this study, we present a method to filter old outliers from age distributions using constant criteria. The algorithm removes
older outlier dates from an age distribution which are separated from the rest of the dataset by a fixed relative time gap. Because
old outliers introduce low gradient regions on an ECDF (e.g. black curve on Fig. 2), we identify potential antecrysts based on
the gradient of an ECDF (Fig. 1C). The model first calculates the gradient of an ECDF, where $\hat{F}_n(t_{rel})$ is the interpolated
ECDF:

$$\nabla\hat{F}_n(t_{rel}) = \frac{\mathrm{d}\hat{F}_n(t_{rel})}{\mathrm{d}t_{rel}} \tag{2}$$

A gradient cut-off term ($\nabla_M$) is then defined, that represents the gradient below which a segment of an ECDF will be defined
as marking a discontinuity in the age distribution (Fig. 2). For a scaled age distribution, we then sum the length of the flat
segments of the ECDF older than the two youngest dates (to ignore age gaps at the young end of the distribution which are not
considered here):

$$t_{flat} = \sum_{i=3}^{N-1} \begin{cases} t_{i+1} - t_i & \text{if } \nabla\hat{F}_n(t_i) < \nabla_M \\ 0 & \text{otherwise.} \end{cases} \tag{3}$$

This metric $t_{flat}$ then provides a quantification of the scaled age duration that is not continuously covered with U-Pb dates
(Fig. 2). For age distributions with two discontinuous age populations, $t_{flat}$ will scale with increasing temporal distance
between the two populations. The choice of $t_{flat}$ that is deemed acceptable ($t_{flat_{max}}$) for such an age distribution is subjective
(i.e. at a low $t_{flat}$ the two age populations will be considered continuous).

If an age distribution yields $t_{flat} < t_{flat_{max}}$ it is deemed to come from one continuous age population. In the opposite case
(i.e. $t_{flat} > t_{flat_{max}}$), the oldest date is iteratively removed until a continuous age population ($t_{flat} < t_{flat_{max}}$) is obtained.
The two parameters $\nabla_M$ and $t_{flat_{max}}$ were tuned until the filtering method was satisfactory in discarding significantly older
zircons throughout the data compilation and did not filter those which were potentially part of the youngest age population. The





optimal parameters determined for $\nabla_M$ and $t_{flat_{max}}$ are 0.30 and 0.25, respectively. Our method has functionality to perform filtering on weighted ECDFs (i.e. taking into account analytical uncertainty). However, we do not implement the filtering on weighted ECDFs because it proved challenging to select constant parameters that would filter all datasets to an acceptable degree. We emphasise that our approach does not aim to provide a geologically significant method to isolate autocrystic zircons (which is impossible to verify), but rather a systematic method of isolating the youngest, continuous population across multiple datasets.

## 2.3   Igneous zircon geochronology compilation

Making interpretations about the dynamics of magma reservoirs from zircon age distributions requires confidence that the dispersion exhibited by the dataset is predominantly geological rather than analytical. One requirement is a sufficiently long duration of zircon crystallisation in a sample relative to the average analytical uncertainty ($\Delta t/\sigma$, as used by Keller et al., 2018). A high $\Delta t/\sigma$ allows the shape of a distribution to be deconvolved from analytical uncertainties, whereas low $\Delta t/\sigma$ age distributions are dominated by analytical uncertainty. Sufficient $\Delta t/\sigma$ in individual magmatic systems is generally only achieved by two analytical techniques: $^{230}$Th–$^{238}$U disequilibrium dating applied to young zircon (predominantly obtained with *in situ* methods such as secondary ion mass spectrometry – SIMS or laser ablation inductively coupled plasma mass spectrometry – LA-ICPMS) and high-precision U–Pb geochronology by (chemical abrasion) isotope dilution thermal ionisation mass spectrometry – (CA-)ID-TIMS (Schaltegger et al., 2015; Guillong et al., 2016). The $^{230}$Th–$^{238}$U method achieves variable, per cent-level precision which may be sufficient to resolve age distributions in the young rocks it is best suited to (< ca. 300 ka; Schmitt, 2011). However, difficulty in calculating reliable individual zircon model ages in the absence of a matching coeval melt or other mineral phase, the effective upper age limit of *ca.* 300 ka, variable and asymmetric age uncertainties, and a focus of existing datasets on volcanic rocks limit the utility of $^{230}$Th–$^{238}$U data for our study. On the other hand, CA-ID-TIMS U–Pb geochronology is applied to zircons from *ca.* 100 ka to the age of the Solar System, achieves a typical precision of $^{206}$Pb/$^{238}$U dates between 0.01–0.1%, and is widely applied to magmatic zircon from a variety of settings (Schoene, 2014). Given that the resolving power of U–Pb geochronology decreases with age, we focused our analysis exclusively on CA-ID-TIMS data for the $^{206}$Pb/$^{238}$U chronometer most applicable to young (<1 Ga) rocks.

In order to systematically compare zircon age distributions between different magmatic systems, we adopted a previously compiled database of published igneous zircon U-Pb dates (Markovic et al., 2024). We classified and sub-selected data from samples clearly identifiable as either plutonic, porphyry, or proximal volcanic deposits. This permits a comparison of age distributions in a diverse range of igneous rocks. We excluded distal volcanic materials such as ash beds or bentonites to avoid biases related to transport sorting or admixed detrital material. While complete exclusion of cases of Pb loss is not verifiable, we focused our analysis on samples least affected by radiation damage, only considering datasets with age <130 Ma. Rare cases of clear young outliers remaining in the database were excluded manually.

We only include age distributions with $\Delta t/\sigma$ greater than 10 to provide confidence that the age distribution is sufficiently dispersed to isolate geological dispersion from analytical dispersion. Resolving an age distribution also requires a sufficient number of dates ($n_{zircon}$) from a magmatic unit. Many studies often report a small number of dates (e.g. five or less), and



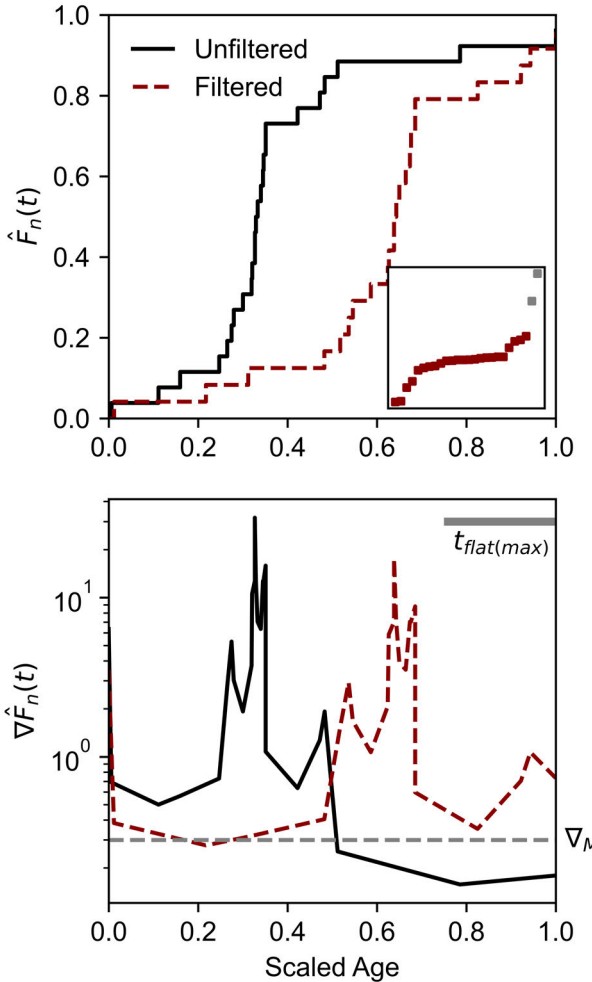

**Figure 2.** An example of the methodology used to filter older age populations from the age distribution of a sample from the Orano dyke swarm, Elba island (Barboni et al., 2015). Upper plot shows the weighted ECDF of the age distribution before and after the filtering process. Inset diagram shows the ranked age plot, with gray symbols highlighting the two dates that were filtered by the method (uncertainties are not shown). Lower plot shows the gradient of the ECDF ($\nabla \hat{F}_n(t)$) before and after filtering. The dashed horizontal grey line indicates the gradient cut-off ($\nabla_M$) below which some degree of inheritance is deemed to have taken place, if the distance in scaled age is greater than the threshold ($t_{flat_{max}}$).

in such scenarios the underlying age distribution is likely undersampled. The final dataset was thus limited to include only inheritance-filtered distributions that contained a minimum of 10 dates, as previous works indicate that sampling with less than 10 zircon dates fails to capture the underlying distribution (Klein and Eddy, 2023; Tavazzani et al., 2023b). We additionally found that some age distributions contain dates with highly variable analytical uncertainties, with some containing over an order

of magnitude variation in uncertainties for zircon crystals dated from the same rock. Individual dates with high uncertainty in





**Table 2.** Sources of data used in the filtered zircon ID-TIMS U–Pb database and metadata describing the type of magmatic emplacement (plutonic, volcanic or porphyry), the mean $1\sigma$ analytical precision and the number of spectra present in the database for each locality.

| Locality | Type | Mean $\Delta t$ (Ma) | Mean $\Delta t/\sigma$ | $n_{spectra}$ | Reference |
|---|---|---|---|---|---|
| Adamello | Plutonic | 0.35 | 14.0 | 1 | Schoene et al. (2012) |
| Agua de Dionisio | Volcanic | 0.13 | 17.9 | 1 | Buret et al. (2017) |
| Bajo de la Alumbrera | Porphyry | 0.16 | 15.3 | 3 | Buret et al. (2016) |
| Batu Hijau | Porphyry | 0.21 | 16.4 | 3 | Large et al. (2020) |
| Bear Valley | Plutonic | 0.58 | 14.4 | 3 | Klein et al. (2021) |
| Bergell | Plutonic | 0.55 | 36.7 | 7 | Samperton et al. (2015) |
| Bingham Canyon | Porphyry | 0.39 | 37.7 | 2 | Large et al. (2021) |
| Capanne | Plutonic | 0.31 | 80.6 | 6 | Barboni et al. (2015) |
| Carpathian-Pannonian | Volcanic | 0.46 | 40.1 | 3 | Brlek et al. (2023) |
| Chegem | Volcanic | 0.08 | 41.9 | 1 | Bindeman et al. (2021) |
| Chuquicamata | Porphyry | 0.73 | 27.2 | 8 | Virmond et al. (2024) |
| Long Valley | Volcanic | 0.03 | 10.0 | 1 | Ickert et al. (2015) |
| Mogollon-Datil | Volcanic, Plutonic | 0.72 | 28.3 | 6 | Rioux et al. (2016); Szymanowski et al. (2019); Gaynor et al. (2023) |
| New England | Plutonic | 2.45 | 37.8 | 2 | Kinney et al. (2021) |
| Ok Tedi | Porphyry, Plutonic | 0.12 | 11.7 | 3 | Large et al. (2018) |
| Radomiro Tomic | Porphyry | 1.19 | 44.2 | 4 | Virmond et al. (2024) |
| Searchlight | Plutonic | 0.16 | 10.9 | 1 | Eddy et al. (2022) |
| Southern Rocky Mtns | Volcanic | 0.99 | 22.4 | 5 | Wotzlaw et al. (2013); Curry et al. (2021) |
| Spence | Porphyry | 0.80 | 13.8 | 4 | Bunker (2020) |
| Toba | Volcanic | 0.27 | 84.3 | 4 | Szymanowski et al. (2023) |
| Turkey Creek | Volcanic | 0.30 | 16.7 | 1 | Deering et al. (2016) |
| Yellowstone | Volcanic | 0.10 | 10.9 | 1 | Wotzlaw et al. (2015) |

an age distribution impact the ability to resolve geological dispersion and in many cases would not be filtered using $\Delta t/\sigma$. We therefore calculate the weighting $w$ that each date $i$ holds in an age distribution using the inverse squared uncertainty (McLean et al., 2011):

$$w_i = \frac{\frac{1}{\sigma^2}}{\sum_i^p \frac{1}{\sigma^2}} \tag{4}$$

Age distributions with a standard deviation of $w_i$ exceeding 0.08 were discarded. The final, filtered compilation contained 70 U-Pb age distributions from 22 magmatic systems (Fig. 3, Fig. S1, Table 2 and Table S1).



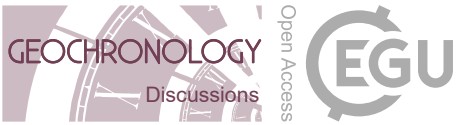

**Figure 3.** Weighted empirical cumulative distribution frequency curves for all 70 filtered zircon age distributions used in this study. Plots are separated based on the locality and curves indicate separate geological units within the locality. The color of each curve reflects the sample classification as volcanic (blue), plutonic (red) and porphyry (yellow) in the database.



## 2.4 The Wasserstein distance

The Wasserstein distance arises from the field of optimal transport and is a metric that allows comparison of two probability distributions. The metric is often termed the "earth mover's distance" because each probability distribution can be treated as a
mound of earth, where the minimum cost of transferring earth from one mound to the other is the amount of earth multiplied by the distance it must be moved (Monge, 1781). Thus, the Wasserstein distance seeks to find the most efficient transportation plan, which is the minimum cost of transporting one distribution to another. The optimal transport plan between two distributions is a measure of the dissimilarity of two distributions, with more dissimilar distributions requiring a greater cost and resulting in a larger Wasserstein distance. For two univariate age distributions, $\mu$ and $\nu$, with cumulative distribution functions (CDFs) M
and N, the $p$th Wasserstein distance between them is given as:

$$W_p(\mu,\nu) = \left( \int_0^1 \left| M^{-1} - N^{-1} \right|^p dt \right)^{\frac{1}{p}} \tag{5}$$

The $W_1$ distance (i.e. where $p = 1$) is equal to the area between two ECDFs (Fig. 4). However, we follow the approach of Lipp and Vermeesch (2023) and implement the Wasserstein-2 distance ($W_2$) which is the squared distance ($p = 2$) and is akin to the standard distance metric used in most statistical analyses. The Python package *Python Optimal Transport* (v. 0.9.4 Flamary
et al., 2021) is used for all optimal transport computations.

Zircon age datasets are discrete data and do not follow a continuous probability distribution and, as such, the Wasserstein distance applies to their empirical cumulative distribution frequency (ECDF) functions. In the case of ID-TIMS datasets, each date is not expected to hold equal weight due to variable analytical uncertainties. Thus the probability distributions $\mu$ and $\nu$, can be represented as the weighted sum of $p$ and $q$ delta functions $\delta$ (Lipp and Vermeesch, 2023):

$$\mu = \sum_i^p m_i \delta_{x_m}, \nu = \sum_i^q n_i \delta_{x_n} \tag{6}$$

where $m$ and $n$ are weights that sum to 1. In the case where dates do not hold equal weight in the overall distribution, weights $m$ and $n$ can be calculated using Equation 4. We plot ECDFs and calculate the $W_2$ distance using the weights, and provide the same results calculated without weights for comparison (Figs. S2 and S3).

The Wasserstein distance has several advantages as a metric to compare zircon age distributions. In addition to allowing
weighting based on uncertainty, it can be used on discrete ages (i.e. an ECDF) and does not require the inference of a specific prior distribution. The $W_2$ is also attractive in that, for two age distributions it is sensitive to their location (their means), spread (their standard deviations) and the shape of the distribution (Irpino and Romano, 2007; Lipp and Vermeesch, 2023). All three of these properties are relevant when considering two U-Pb age distributions. We prefer the Wasserstein distance to the Kolmogorov-Smirnov distance because the latter is a measure of the maximum vertical difference between the two ECDF and
is, as such, less sensitive to the overall shape of the distribution.





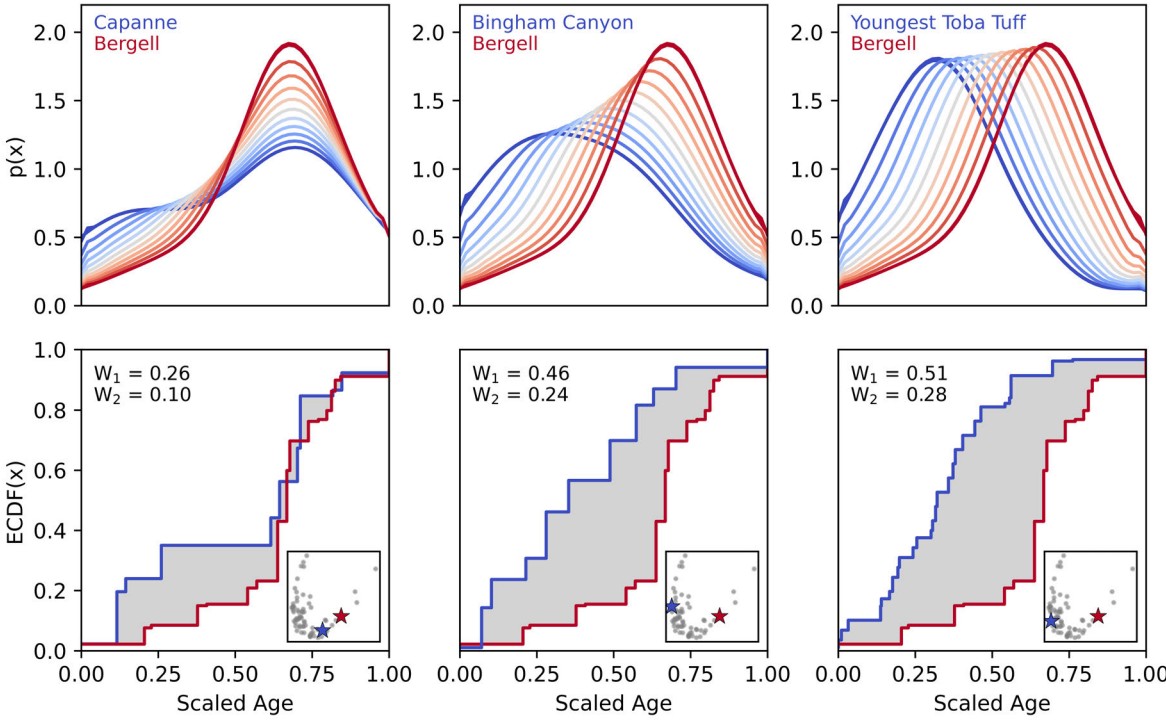

**Figure 4.** Examples showing three comparisons of similar to contrasting U-Pb ID-TIMS age distributions. Top row shows the distributions in blue and red with the transition distributions showing equally spaced (at 0.1 intervals) distributions in Wasserstein space (i.e. the Wasserstein barycentres). Bottom row shows the same two distributions as above as weighted ECDF curves, where the grey shaded area indicates the $W_1$ distance between the two distributions. The $W_2$ is also reported. Insets show where the two compared distributions plot on the PC1 versus PC2 diagram of the $W_2$ dissimilarity matrix (Fig. 5).

## 2.5 Dissimilarity matrix and dimensionality reduction

The $W_2$ metric permits pairwise comparisons of age distributions. Thus for $Y$ age distributions, a symmetric dissimilarity matrix, $d$ of dimension $Y \times Y$ can be constructed:

$$
d = \begin{bmatrix}
d_{1,1} & d_{1,2} & \cdots & d_{1,Y} \\
d_{2,1} & d_{2,2} & \cdots & d_{2,Y} \\
\vdots & \vdots & \ddots & \vdots \\
d_{Y,1} & d_{Y,2} & \cdots & d_{Y,Y}
\end{bmatrix}
\tag{7}
$$

Because the $W_2$ distance is a metric (i.e. it satisfies the triangle inequality), principal component analysis (PCA) can be applied to reduce the dissimilarity matrix to fewer dimensions whilst preserving the pairwise distances (Vermeesch, 2013; Lipp





and Vermeesch, 2023). PCA aims to preserve the variance of a dataset in a lower dimension space, where for a dissimilarity matrix this reflects the pairwise distances. We use the Python implementation within *sci-kit learn* (v. 1.5.1 Pedregosa et al., 2011) for PCA. The first two principal components contain 90% of the variance of the dissimilarity matrix, indicating that the
majority of the pairwise distances are preserved with only 10% being lost.

    Previous studies typically have used multi-dimensional scaling (MDS) to reduce a dissimilarity matrix of age distributions to 2D (Vermeesch, 2013; Lipp and Vermeesch, 2023). However, it cannot be used to cast synthetic data into the same two dimensional space as natural data without re-transformation, and can produce variable results with different seeds. Because a primary aim of our study is to compare the dissimilarity matrix of natural data to synthetic and modelled distributions, PCA
is a more suitable dimensionality reduction technique. A comparison of the PCA and MDS suggests no significant difference between the results of both dimensionality reduction approaches (Fig. S4).

## 2.6 Modelling approach

### 2.6.1 Bootstrap forward modelling of zircon age distributions

Forward modelling of synthetic age distributions and comparison with natural data can yield insights into the controls on age
distributions in magmatic systems (Samperton et al., 2017; Keller et al., 2018; Tavazzani et al., 2023a). This process aims to sample an underlying zircon age distribution according to realistic uncertainties and numbers of zircons sampled for ID-TIMS datasets. The underlying distribution, or the uncertainties and number of zircons sampled can be varied to test different hypotheses on the controls on zircon age distributions.

    We follow the boostrap modelling approach of Tavazzani et al. (2023b). The model samples $n_{zircon}$ synthetic dates from a
probability distribution $p(x)$ which is equivalent to a selected underlying age distribution, such as the theoretical zircon age distribution obtained from zircon solubility and thermodynamic modelling of a monotonically cooling magma reservoir (Keller et al., 2018). Gaussian uncertainty is then added to each synthetic age to reproduce realistic uncertainties in ID-TIMS data. To calculate the uncertainty at a given age, we parameterised the $2\sigma$ absolute analytical uncertainty as a function of $^{206}$Pb/$^{238}$U age ($t$ in Ma) for the entire ID-TIMS compilation (Markovic et al., 2024) using a second order polynomial fit. The resulting
best fit parameters and covariance matrix yields the following equation with errors given as 1*SE*:

$$2\sigma = 5.5 \times 10^{-7}(\pm1.4 \times 10^{-7})t^2 + 8.9 \times 10^{-4}(\pm1.0 \times 10^{-4})t + 0.040(\pm0.0081) \tag{8}$$

    Gaussian uncertainty is propagated onto each date sampled during boostrap sampling according to the standard error on the fit parameters. The bootstrap sampling can be repeated with varying numbers of zircon grains, different age (and thus different analytical uncertainties) and with a different underlying distribution. These distributions can then be concatenated
with the $W_2$ dissimilarity matrix generated on natural data. The pairwise dissimilarities of a modelled distribution with all natural distributions can then be transformed by PCA and visualised alongside natural data.





### 2.6.2   Magma recharge and underlying age distributions

The underlying age distribution from which zircon U-Pb dates are sampled can vary as a function of magmatic flux and
volume (Caricchi et al., 2014; Tavazzani et al., 2023a). Samperton et al. (2017) showed a remarkable similarity between age
distributions predicted from zircon solubility (Watson, 1996) and thermodynamic models (Keller et al., 2018), and those dated
by ID-TIMS in intrusive rocks. This predicts a peak in zircon crystallisation at the onset of zircon saturation which decays until
the solidus, producing a skew towards older ages. However, this prediction assumes monotonous cooling of magma, while
open-system behaviour can produce multi-modal age distributions with a general shift towards younger skew (Schmitt et al.,
2023; Tavazzani et al., 2023a).

235       We use the approach of Tavazzani et al. (2023b) who generated zircon age distributions representative of complex crystalli-
sation simulations. These simulations investigate zircon crystallisation under a range of non-linear temperature-crystallinity
scenarios developed using the thermodynamic modelling software Magma Chamber Simulation (Bohrson et al., 2014). This
combines closed- and open-system processes such as fractional crystallisation, single or repeated recharges of new magma in a
magma body. Synthetic age distributions generated under varying recharge can then be compared to natural data as described
above. We compare the bootstrap sampling of a zircon age distribution generated with recharge of a cooling, upper crustal rhy-
olitic magma reservoir with 0, 3 and 5 recharges. Each recharge is triggered by crystallinity reaching 50 vol.% and comprises
an addition of 50 g of rhyolitic magma (with an initial magma reservoir mass of 100 g) with the same composition and liquidus
temperature (870°C) as the original magma (Tavazzani et al., 2023b).

## 3   Results and Discussion

### 3.1   Dissimilarity between plutonic, volcanic and porphyry zircon age spectra

The PCA plot of the $W_2$ dissimilarity matrix of the high precision U-Pb data compilation produces a parabolic shape (Figure
5). Because the $W_2$ distance is equivalent to the squared difference between two weighted ECDFs, the location of an age distri-
bution on the plot of the PCA of the $W_2$ dissimilarity matrix (Fig. 5) can be compared with the shapes of the weighted ECDFs
of the age distributions (Fig. 3). The distance of two age distributions on the PCA plot refers to the degree of dissimilarity
between them. For example, two strongly contrasting distributions (Fig. 3) such as from a New England plutonic sample (old
skew) and a Yellowstone volcanic sample (young skew) plot on opposite sides of the PCA plot (Fig. 5). By contrast, relatively
homogeneous weighted ECDFs in single magmatic systems cluster together on the PCA plot (e.g. Toba).

There is an overall divide between volcanic and plutonic systems with porphyry age distributions appearing more similar to
volcanics (Fig. 5). In order to interpret this diagram, we cast skew normal distributions onto the $W_2$ dissimilarity matrix which
were generated with varying values of skew (see inset plots in Fig. 5). This comparison demonstrates that PC1 generally reflects
the amount of negative (old) skew of a distribution, because increasingly positive PC1 scores show increasing skew to older
ages, whilst young-skewed and non-skewed distributions have PC1 scores close to zero. In turn, PC2 correlates with absolute
values of skew, with higher values indicating a greater deviation from a normal distribution and negative values indicating a





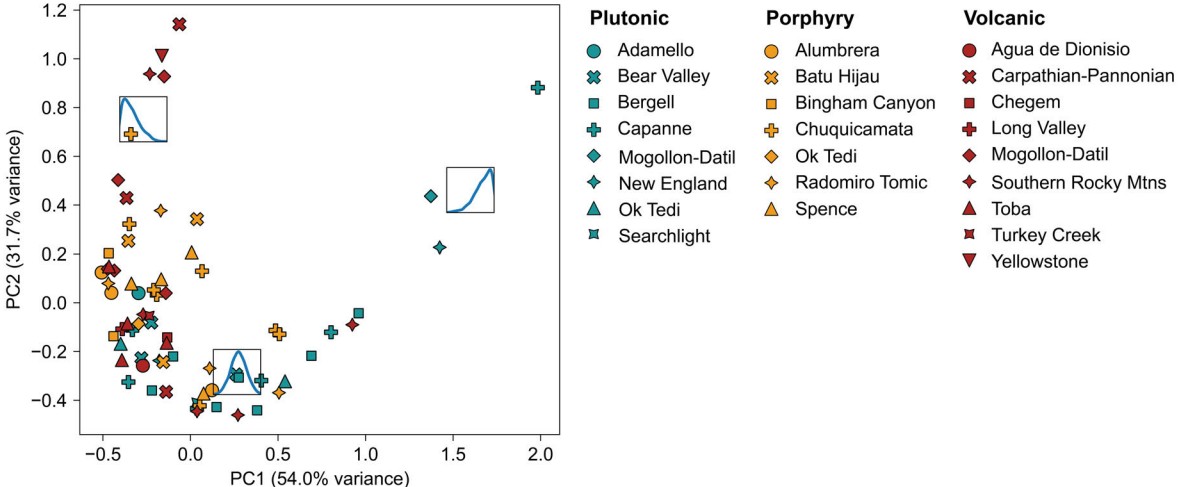

**Figure 5.** Results of PCA on the $W_2$ ID-TIMS zircon U-Pb dissimilarity matrix. Each data point indicates one U-Pb age distribution for a magmatic system. The percentage variance that each principal component accounts for is given. Three inset graphs represent KDEs of skew normal distribution with varying values of skew (-100, 0 and 100) for comparison with natural distributions.

normal distribution. Volcanic and porphyry age distributions generally have low PC1 scores indicating a predominant skew
towards younger ages in their distributions. By contrast, plutonic age distributions have more variable PC1 scores but a large population have high PC1 scores indicating skew towards older ages.

### 3.2  Controlling factors on age spectra

#### 3.2.1  Analytical factors

As discussed previously, the ability to resolve geological dispersion from an age distribution depends strongly on $\Delta t/\sigma$.
The selected threshold of $\Delta t/\sigma$ permitted in our database ($> 10$) aims to reduce the number of age distributions which are dominated by analytical uncertainty. In order to validate our choice of $\Delta t/\sigma$ filter, we examined the position of distributions on the PCA of the $W_2$ dissimilarity matrix relative to their $\Delta t/\sigma$ (Figure 6A). We do not find that low values of $\Delta t/\sigma$ (i.e. close to 10) lead to a collapse of distributions towards normal distributions, indicating that these distributions are still dominated by geological dispersion. We additionally find that repeating our analysing with a lower threshold of $\Delta t/\sigma$ ($> 5$) provides
similar results (Fig. S5) yet further populates the central part of the PCA of the $W_2$ dissimilarity matrix, potentially indicating these lower $\Delta t/\sigma$ distributions have less skew and may be more dominated by analytical uncertainty. A significant number of distributions also yield very large $\Delta t/\sigma$ (e.g., $> 80$).

We limited our study to ID-TIMS datasets to young datasets ($< 130$) and where chemical abrasion was performed to reduce the affects of Pb loss. However, we find that the conditions under which chemical abrasion was performed vary. Notably,
the temperatures of chemical abrasion range between 180 and 220°C which can impact the measured U-Pb date (Widmann




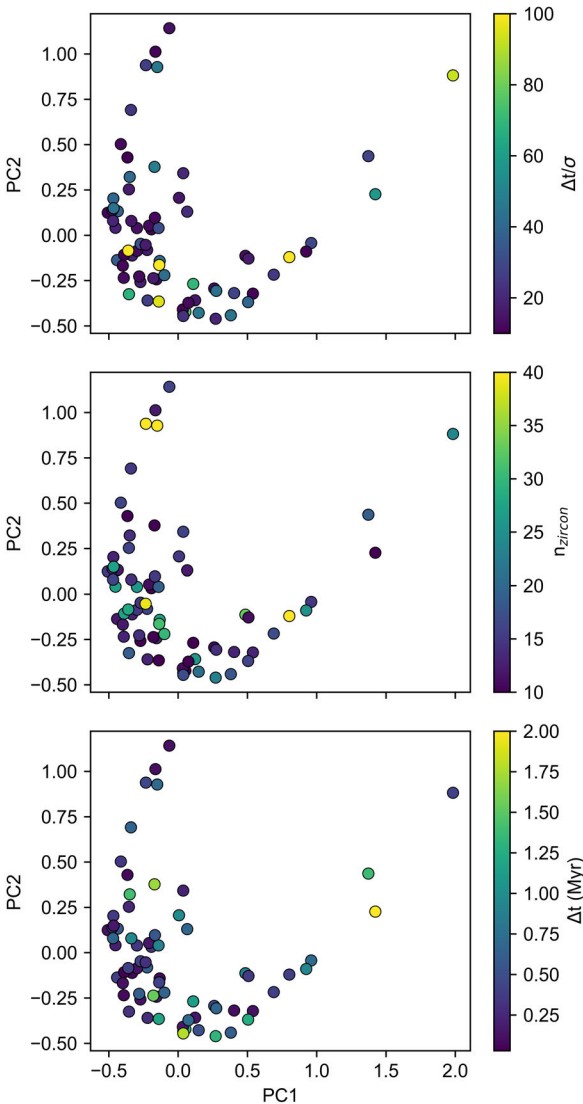

**Figure 6.** Plot of the PCA on the $W_2$ ID-TIMS zircon U-Pb dissimilarity matrix coloured by: (A) the ratio of zircon crystallisation duration and the mean analytical uncertainty ($\Delta t/\sigma$), (B) the number of zircons in the age spectrum ($n_{zircon}$) and (C) the duration of zircon crystallisation in Myr ($\Delta t$).

et al., 2019).To exclude the possibility that the variability in shape of an age distribution could be the result of unmitigated Pb loss, we compiled the chemical abrasion temperature of zircon in each study, and compared this with the position of each distribution on the PCA plot of the $W_2$ dissimilarity matrix (Fig. S6). If unmitigated Pb loss is present, we should find that age distributions from lower chemical abrasion temperatures have higher PC1 and PC2 scores. However, we find no relationship




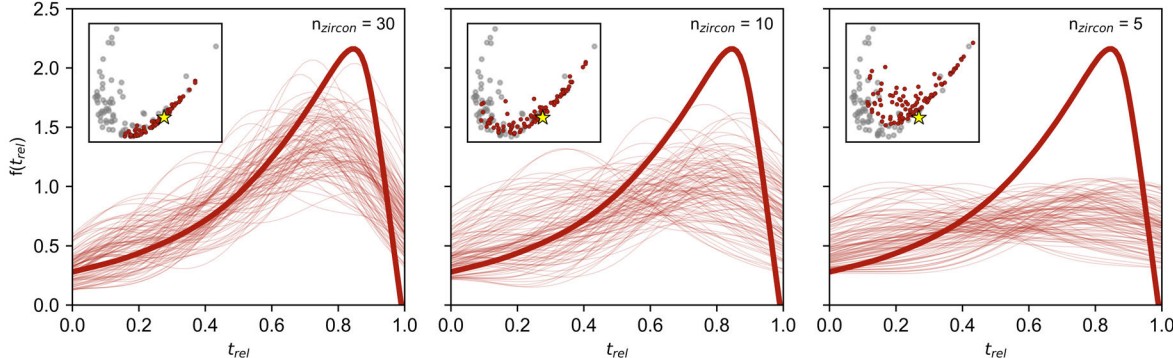

**Figure 7.** Bootstrap sampling of the theoretical zircon crystallisation distribution in a monotonically cooling magma reservoir derived from thermodynamic models (Keller et al., 2018) as shown by the thick red curve. The number of zircons sampled ($n_{zircon}$) ranges from 30 (left), 10 (centre) to 5 (right) and the thin translucent red curves show 100 iterations of sampling. An uncertainty is added to each age as appropriate for a sample at 10 Ma. Inset plots show the PCA of the $W_2$ dissimilarity matrix of natural zircon U-Pb distributions (grey symbols) and where the 100 synthetic age distributions plot (red symbols).

between chemical abrasion temperature and distribution shape, which excludes the possibility that the varying shape of age distributions in CA-ID-TIMS datasets is related to unmitigated Pb loss.

### 3.2.2   Number of zircons

A decreasing number of zircon analyses ($n_{zircon}$) in an age distribution leads to a collapse of an age distribution towards a normal distribution (Tavazzani et al., 2023a). To further test the validity of our choice of $n_{zircon} > 10$ as a filter, we systematically
study the effect of $n_{zircon}$ by sampling different numbers of zircon from the same underlying synthetic zircon crystallization distribution in a monotonically cooling magma reservoir derived from thermodynamic models (for a full description of the methodology used to obtain this theoretical distribution see Keller et al. (2018); Tavazzani et al. (2023b)). We perform this sampling for $n_{zircon}$ of 5, 10 and 30 at analytical uncertainties typical for 10 Ma (Fig. 7) and provide a more extensive analysis at different ages (Fig. S7) and for $n_{zircon}$ between 4 and 60 (Fig. S8). Our search space for $n_{zircon}$ broadly reflects the lower,
intermediate and upper range of zircon analysed per sample in the compiled ID-TIMS datasets (Table S1).

Results of iterative sampling from an underlying monotonic cooling distribution highlight the loss of reproducibility of the initial distribution when $n_{zircon} = 5$ (Fig. 7). This is reflected by the increasing flattening of the KDE curve and, in PCA space of the $W_2$ dissimilarity matrix, by a shift of synthetic zircon distributions towards lower PC1 scores. Undersampling of the underlying distribution can lead to a reversal of the skew in the bootstrapped distribution (i.e. negative PC1 scores; Fig. 7).
This confirms that the shape of age distributions in datasets where $n_{zircon} < 10$ should not be interpreted in the context of geological variability.





Sampling a larger number of zircons ($n_{zircon}$ = 10) more accurately captures the skew towards older ages of the underlying distribution (Fig. 7). However, an appreciable variation is still apparent in these sampled distributions which cover a large range of PC1 scores in the $W_2$ dissimilarity matrix. With an even larger number of sampled zircon dates ($n_{zircon}$ = 30), the

variance of the sampled distributions reduces as they more accurately reproduce the underlying distribution. However, even in the case where the number of sampled zircon dates is high (n = 30), significant variation still exists in the PCA plot of the $W_2$ dissimilarity matrix (Fig. 7). This finding suggests one should avoid making interpretations from small variations between different spectra on the PCA plot of the $W_2$ dissimilarity matrix, since these are likely to reflect undersampling.

More detailed modelling of the effect of $n_{zircon}$ on the ability to capture an underlying age distribution (Fig. S8) shows that

at 10 Ma, the optimal choice of $n_{zircon}$ is approximately 14 because increasing $n_{zircon}$ beyond this leads to an only marginal increase in performance. At 100 Ma, producing an equivalent performance requires approximately 20 zircon grains to be dated. However, our choice of $n_{zircon}$ = 10 for this study can still capture an underlying distribution and a greater threshold would significantly impact the availability of natural data for this study. As larger datasets are published, the filtering threshold of $n_{zircon}$ could be increased to further reduce the effect of undersampling on the shape of the age distribution.

### 3.2.3  Truncation

Zircon age distributions in volcanic and porphyry rocks are produced by a characteristic crystallisation history, because magma quenching upon evacuation/eruption after zircon saturation will interrupt zircon crystallisation (Keller et al., 2018; Schmitt et al., 2023), a process we refer to herein as "truncation" of the age spectrum. We model the effect that truncation has on the resulting zircon age spectra by using the same underlying monotonic cooling zircon crystallisation distribution used above,

sampled at variable degrees of $t_{rel}$, namely 0.8, 0.5 and 0.1. This simulates scenarios where a monotonically cooling igneous body erupts at a time close to the solidus relative to zircon saturation ($t_{rel}$ = 0.8), halfway in time between zircon saturation and the solidus ($t_{rel}$ = 0.5) or close in time to zircon saturation ($t_{rel}$ = 0.1). We model this range as volcanic products can vary between crystal-poor and crystal-rich (Hildreth, 1981) and porphyry rocks are typically crystal-rich, though we emphasise that $t_{start}$ refers to the time of zircon saturation and not the liquidus. The simplified model assumes that melt evacuation quenches

the melt and abruptly terminates zircon crystallisation.

The zircon age distribution of a magma extracted at $t_{rel}$ = 0.8 are comparable with the distribution obtained from a monotonically cooled magma body, although some of the skew toward older ages is lost (Fig. 8). These sampled synthetic distributions produced by mild truncation of the underlying monotonic cooling distribution reproduce the spread of the natural dataset on the right arm of the PCA plot of the $W_2$ dissimilarity matrix. When truncation takes place at an intermediate time between

zircon saturation and the system's solidus ($t_{rel}$ = 0.5), the effect of truncation of the underlying monotonic cooling distribution is more evident. In this scenario, synthetic zircon age spectra lose most of their skew and largely overlap with the central axis of the natural zircon U-Pb distributions in the PCA space of the $W_2$ dissimilarity matrix (i.e. form a near-normal distribution). In the case of truncation happening at $t_{rel}$ = 0.1, synthetic age distributions always exhibit negative skew (i.e. low PC1 scores). In the PCA space of the $W_2$ dissimilarity matrix, these do not overlap at all with the underlying monotonic cooling zircon age

distribution.

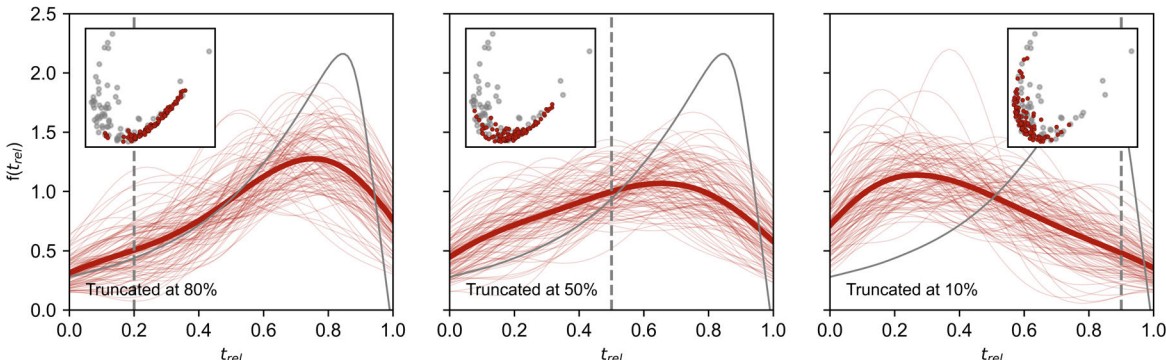

**Figure 8.** The effect of sampling an underlying distribution with different degrees of truncation of zircon crystallisation ($t_{rel}$ = 0.8; 0.5; 0.1) simulating melt extraction/volcanic eruption. The sampled underlying distribution (grey dashed line) is the monotonic cooling distribution (Keller et al., 2018). Thirty zircons are sampled for each of the 100 synthetic distribution (faint red) and the average of 100 simulations is shown (dark red). Inset plots show the location of these distributions (red symbols) in the PCA plot of the $W_2$ dissimilarity matrix of natural zircon U-Pb distributions (grey symbols).

These results indicate that truncation can significantly alter the shape of the underlying distribution, with extreme truncation ($t_{rel}$ = 0.1 in our simplified model) being able to reverse the skew of a distribution. Our modelling also shows that truncation can account for most of the variability in zircon age distributions observed in natural examples. Therefore, this likely is a key factor in explaining the younger skew of volcanic and porphyry zircon age distributions relative to plutonic zircons, given the

former age distributions will be interrupted by magma evacuation and subsequent rapid cooling. We note however that even the most extreme effects of truncation are not able to reproduce a minority of volcanic zircon age spectra nested on the far top-left end of the $W_2$ dissimilarity matrix curve (Fig. 8).

### 3.2.4 Magma recharge

It has been suggested previously that differences in zircon age distributions between volcanic and plutonic systems may be

linked to different fluxes in the source magma reservoir (Caricchi et al., 2014; Schmitt et al., 2023). The forward crystallization models investigated so far assume a monotonically cooling magma reservoir with no subsequent recharge (Figs. 7 and 8), which may be unrealistic for many magmatic systems (Sparks et al., 1977), where recharge will increase temperature, change melt composition, and thus influence zircon crystallisation in the magmatic system (Szymanowski et al., 2020; Tavazzani et al., 2023a). We therefore consider the role of sampling different underlying distributions of zircon ages whose shape may be

produced by different magma recharge histories. In these scenarios, the compositional and thermal contribution of each new magma injection acts against zircon saturation and consequently delays zircon crystallisation.

We investigate the effects of an increasing number of rhyolitic melt recharges (0, 3 and 5 recharges of 870 °C) into a cooling rhyolitic magma reservoir. The underlying distributions transition from unimodal with no recharge, to increasingly multi-modal



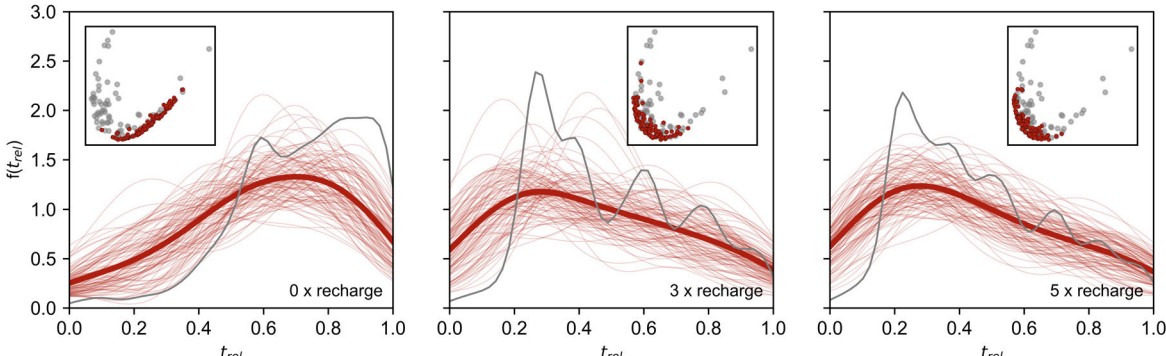

**Figure 9.** The effect of repeated magma recharge on zircon age distributions. Underlying zircon age distributions produced with varying numbers of recharges of felsic melt into a felsic magma reservoir undergoing cooling and crystallisation. Underlying zircon age distributions (grey curves) are taken from Tavazzani et al. (2023b) for (A) 0, (B) 3, and (C) 5 recharge events during magma cooling. Thirty zircons are sampled from 100 sampled distributions and uncertainties are added as appropriate for 10 Ma (red curves). The average curve (thick opaque red) shows the average of the 100 synthetic age distributions and the progressive shift from old to young skew with increasing number of recharges. Insets show the location of these distributions (red symbols) in the PCA plot of the $W_2$ dissimilarity matrix of natural zircon U-Pb distributions (grey symbols).

and with younger skew as the number of recharges increases (Fig. 9). With no recharge, the synthetic zircon crystallisation
spectrum is equivalent to the monotonic cooling distribution and sampled distributions all exhibit older skew and are located on the right hand side of the PCA plot. With three recharges, the synthetic distributions have predominantly younger skew and plot towards the centre and left of the PCA plot (Fig.9). With five recharges, the synthetic distributions look broadly similar to those after three recharges, but with less variance and a slight increase in the skew towards younger ages (Fig.9). Therefore, recharge appears to exert a strong control on the skew of zircon age distributions, in a similar albiet less extreme manner to
truncation of zircon crystallisation. Furthermore, like truncation, it is unable to produce the age distributions plotting at the lowest PC1 and highest PC2.

### 3.2.5  "Antecrysts" in zircon age distributions

The natural age distributions that plot at the lowest PC1 and highest PC2 are those with a minority of zircon dates that are significantly older than the main population. Because these age distributions can't be reproduced by the modelling performed
so far, it is possible these distributions contain "antecrysts" (Miller et al., 2007) that are only marginally older than the broader age distribution (and hence not identified by our filtering method).

    The modelling presented so far samples an age distribution from a single magma reservoir. To model the effect of "antecrysts" on zircon age distributions, we additionally sample $n_{inh}$ zircons from a second age distribution which is older than the primary age distribution in relative time $\Delta t_{rel}$. The population of $n_{inh}$ is then added to a primary bootstrapped age distribution




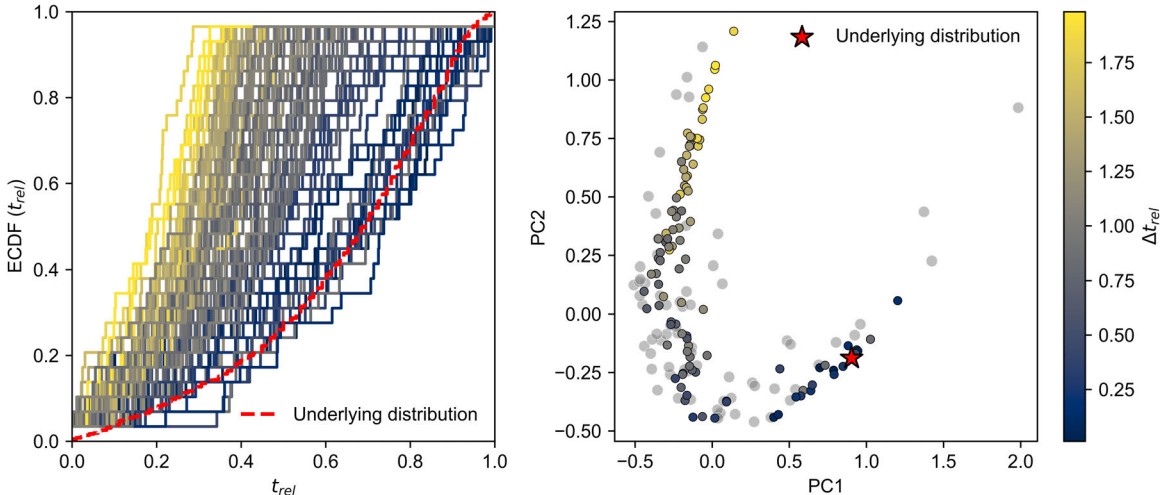

**Figure 10.** The effect of antecrysts on zircon age distributions for 100 simulations where 29 zircons are sampled from a primary age distribution and one zircon is sampled from an identical age distribution older by $\Delta t_{rel}$ from the primary distribution. $\Delta t_{rel}$ is randomly varied between 0 and 2 and each synthetic distribution is coloured by the magnitude of $\Delta t_{rel}$. (A) The ECDF of each synthetic distribution and (B) the location of each synthetic distribution in the PCA plot of the $W_2$ dissimilarity matrix of natural zircon U–Pb distributions.

365 consisting of $n_{zircon} - n_{inh}$ zircons and then $t_{rel}$ is renormalised between zero and one. These modelled age distributions can then be treated identically to those modelled above, and compared with natural data in the PCA $W_2$ space.

 Our modelling of the effect of sampling antecrysts on age distributions shows that the presence of just one antecrystic zircon in an autocrystic population of 30 zircon U-Pb dates can have a profound effect on the final age distribution (Fig. 10). An antecryst derived from an older secondary age distribution where $\Delta t_{rel} = 1$ (i.e. zircon crystallisation in older distribution

370 ended at the same time as zircon saturation in the primary, younger distribution) has a discernible effect on the age distribution, shifting it from plotting alongside older skewed datasets to where those with normal distributions plot (Fig. 10). Larger values of $\Delta t_{rel}$ (up to two) have an even greater effect, and can produce the lowest PC1 and highest PC2 scores observed in the natural data. Thus in summary, the natural zircon age distributions with the most extreme young skew appear to be best explained by the incorporation of antecrystic zircon.

## 375 4 Implications

### 4.1 Geological controls on variations in zircon age spectra

The contrasting age distributions of plutonic, porphyry and volcanic zircons indicate a difference in the magmatic history they record, consistent with comparisons of zircon age distributions in previous studies (Caricchi et al., 2014; Tavazzani et al., 2023a). We have shown that a number of geological factors can impact the shape of an age distribution and the younger skew





of volcanic and porphyry zircon age distributions can be both explained by truncation or magma recharge shifting the peak of zircon crystallisation towards younger ages. Our modelling suggests that truncation has a slightly more extreme effect than magma recharge in producing young skew in age distributions. Though not mutually exclusive, we suggest that truncation is the most likely candidate to explain the difference between volcanic, porphyry and plutonic zircon age distributions because it is a factor that affects all volcanic and porphyry zircon age distributions and does not affect plutonic distributions. We further

found that both truncation and recharge cannot skew an age distribution to the most extreme young skew observed. Instead, this can be attributed to the incorporation of "antecrysts" in an age distribution. In reality, the observed zircon age spectra can be simultaneously affected by many of the geological and analytical effects discussed here, and the deconvolution of their exact contributions will be challenging.

### 4.2 Analytical resolution of zircon age spectra

The effect of geological processes versus analytical uncertainty on an age distribution will vary as a function of sample age (i.e. as absolute analytical uncertainties expand). The interpretations made in this study are based on young (*ca.* < 130 Ma) magmatic systems analysed by ID-TIMS, where analytical capabilities are strongest and the potential for Pb loss is lowest. However, it is also possible that such an approach could be applicable to *in situ* techniques such as LA-ICP-MS for samples young enough that the large relative analytical uncertainties correspond to sufficiently low absolute uncertainties.

We adapt our bootstrap model of the monotonic cooling age distribution to sample age distributions at a wider range of ages (between 0.1 and 1000 Ma) for absolute analytical uncertainties relevant to ID-TIMS and LA-ICP-MS at the given age. This uses the parameterisation of ID-TIMS uncertainties as a function of age (Equation 8) to allow the bootstrap modelling of the monotonic cooling distribution to be performed from 0.1 to 1000 Ma. We also performed a third-order polynomial regression on $2\sigma$ absolute uncertainties reported for a compilation (Chelle-Michou and Schaltegger, 2023) of LA-ICP-MS dates (*t* in Ma)

with uncertainties reported as 1*SE*:

$$2\sigma = -1.8 \times 10^{-9}(\pm 1.9 \times 10^{-10})t^3 + 4.8 \times 10^{-6}(\pm 8.6 \times 10^{-7})t^2 + 0.15(\pm 0.01)t + 0.23(\pm 0.18) \tag{9}$$

Gaussian uncertainty is propagated onto each date sampled during bootstrap sampling according to the standard error on the fit parameters. We perform bootstrap sampling 25 times and calculate the Wasserstein distance between the sampled age distribution and the original age distribution which we refer to as the age distribution "misfit" (Figs. 11, and S9). We average

and plot these 25 misfit scores along with their standard deviation. When the misfit is 0, the bootstrap sampled age distribution fully reproduces the underlying distribution. As the bootstrapped distribution deviates further from the underlying distribution, the misfit increases.

When 10 zircon crystals are sampled, this analysis indicates that at young ages (<5 Ma), LA-ICP-MS and ID-TIMS can capture the underlying distribution to a similar degree. When 30 zircon crystals are sampled, the overlap of analytical perfor-

mance between LA-ICP-MS and ID-TIMS is only present at very young ages (<1 Ma). This indicates that for the youngest systems, it may be possible to resolve a zircon age distribution using LA-ICP-MS where high $n_{zircon}$ datasets can be more

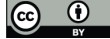


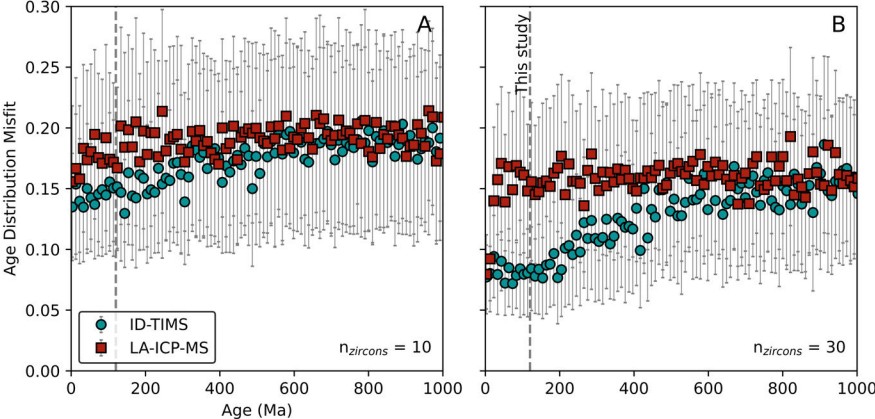

**Figure 11.** Comparison of the ability of ID-TIMS and LA-ICP-MS to capture the underlying zircon age distribution. At a given age, bootstrap forward modelling is applied on the monotonic cooling age distribution (Keller et al., 2018) and the Wasserstein distance is calculated between the sampled age distribution and the underlying distribution. At each age this is repeated 25 times and the mean $W_2$ is shown with the error bars showing the standard deviation. Vertical dashed line shows the maximum age of ID-TIMS distributions permitted in this study. This method is repeated for A) $n_{zircon} = 10$ and B) $n_{zircon} = 30$.

easily acquired, though we emphasise this requires further study since reported LA-ICP-MS analytical uncertainties may be underestimated and often neglect the in-session reproducibility of reference materials (Large et al., 2020; Sliwinski et al., 2022). With increasing age, the absolute precision of LA-ICP-MS decreases linearly, as evidenced by the rapid increase in misfit be-

tween 0 and *ca.* 100 Ma for $n_{zircon} = 10$ . For ages older than this, the misfit remains unchanged with increasing age because analytical uncertainties dominate the sampled distribution, forming a normal distribution. By contrast, for ID-TIMS, the misfit increases less rapidly with age. This continues until approximately 500 Ma, when the misfit is equivalent to LA-ICP-MS and the sampled dates form a normal distribution regardless of the shape of the underlying distribution.

### 4.3   Estimation of eruption/solidification ages

The most common application of zircon age spectra is to calculate $t_{end}$, i.e. the age of a volcanic eruption or emplacement of an intrusion (Keller et al., 2018; Ratschbacher et al., 2018). Although many authors often use the youngest measured date or the weighted mean of the youngest population of dates, Keller et al. (2018) demonstrated that a Bayesian method frequently provides the most accurate result and is least likely to underestimate uncertainty.

The Bayesian approach uses prior knowledge of the zircon crystallisation spectrum (e.g. from theoretical constraints) and

measured zircon U-Pb age data and their uncertainties to estimate $t_{end}$ and its uncertainty. Several options exist for prior distributions such as the monotonic cooling distribution, a uniform distribution or a bootstrapped sample of the measured U-Pb dates. In some cases, the choice of prior may not greatly influence the eruption age (Keller et al., 2018), but in cases where the data are under-dispersed, or analytical uncertainties are high or the number of zircons is low, greater importance is placed on a



prior knowledge of the zircon age distribution. Therefore, our analysis of natural zircon age distributions from young systems
can inform the choice of a prior distribution. Our study unambiguously shows that a monotonic cooling distribution (skew
towards old ages) can be assumed for many plutonic systems but is not appropriate for volcanic or porphyry systems (Fig. 5).
For volcanic and porphyry systems, a truncated distribution appears to be the most appropriate choice of prior (e.g. truncated
normal, half normal or triangular distributions; Keller (2018)) as validated by our modelling of the effect of truncation on
shifting zircon age spectra towards a younger skew (Fig. 8).

## 4.4 Modelling of magma dynamics using zircon age spectra

Zircon U-Pb age spectra provide the most robust time-resolved history of magmatic systems. Several studies have used these
spectra to quantify the dynamics of magma reservoirs, such as the magmatic flux and the duration of magmatism (Caricchi
et al., 2014; Tierney et al., 2016; Weber et al., 2020; Schmitt et al., 2023). Though initially such models assumed linear zircon
crystallisation after zircon saturation (Caricchi et al., 2014, 2016), some recent models have taken into account the thermal
and compositional dependence of zirconium solubility in a silicic melt and its effect on zircon saturation (Tierney et al., 2016;
Schmitt et al., 2023). Modelling indicates that zircon age spectra modelled during open-system magmatic recharge exhibits
a dominance of younger zircon ages (Caricchi et al., 2014; Schmitt et al., 2023; Tavazzani et al., 2023a, this study) and thus
contrasts from the classic zircon age distribution predicted from a monotonically cooling magma reservoir (i.e. the monotonic
cooling distribution; Watson, 1996).
Previous studies based on a limited dataset have also noticed the young-skewed zircon age distributions in volcanic systems
relative to plutonic systems (Caricchi et al., 2014; Tavazzani et al., 2023a). This has been used to suggest volcanic systems
have non-linear thermal histories and overall higher magma recharge rates (Caricchi et al., 2014; Tierney et al., 2016; Schmitt
et al., 2023; Tavazzani et al., 2023a). However, our study indicates that these young skewed zircon age spectra can also be
produced by the truncation of zircon crystallisation by a volcanic eruption. Thus we emphasise that decoupling the effects of
recharge and truncation is challenging, and magmatic fluxes inverted from volcanic age distributions should be treated with
caution, particularly when contrasted with plutonic age distributions. However, between plutonic systems (where truncation is
not a large factor) it may be possible to more robustly quantify magma recharge rates using zircon age spectra. Future work on
retrieving information on magma dynamics from zircon age spectra may therefore be best suited to younger plutonic systems
(e.g. Barboni et al., 2015; Farina et al., 2024).

## 5   Conclusions

We provide a quantitative framework to compare high precision zircon U-Pb age distributions of igneous samples using the
Wasserstein distance, and show that this statistical measure accurately captures differences in shape between age distributions.
Our study provides a comprehensive analysis of a large database of CA-ID-TIMS data from 30 different magmatic systems. We
filter our data compilation to isolate datasets where relative uncertainties permit resolution of geological dispersion. The filter-
ing approach includes a new method to systematically filter the tails of age distributions in large datasets, which we provide for

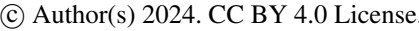

use in the community. Exploring the effect of geological processes on age distributions can be made challenging by varying ratios of zircon crystallisation duration to analytical precision ($\Delta t/\sigma$) and varying numbers of sampled zircons. Notwithstanding these limitations, our analysis of natural data indicates clear geological differences between volcanic, porphyry and plutonic age distributions where volcanic and porphyry zircon age distributions exhibit younger skew and plutonic exhibit older skew. 465 We adopt a bootstrap modelling approach which generates synthetic distributions and permits comparison with natural distributions according to the Wasserstein distance. This analysis indicates that the younger skew of volcanic and porphyry age distributions can be best explained by variable degree of truncation of zircon crystallisation by a volcanic eruption or porphyry dyke emplacement. We also show that magma recharge can contribute towards the younger skew of volcanic and porphyry age distributions, though we suggest that the role of truncation is most important. This major control of zircon crystallisation 470 truncation on an age distribution suggests that interpreting magma dynamics (e.g. quantifying magmatic fluxes) from volcanic age distributions may be problematic, and such an approach is more appropriate for plutonic age distributions. As analytical precision improves and increasing numbers and sizes of CA-ID-TIMS zircon U-Pb datasets are published, our framework will facilitate improved interpretation of geological information from zircon age distributions.

*Code and data availability.* The code to reproduce the findings of this study is deposited in open-source Zenodo repository 475 (https://doi.org/10.5281/zenodo.13378412). The database of zircon age distributions are provided in the Supplementary Data (Table S1) and are deposited in the Zenodo repository.

*Author contributions.* Conceptualisation: CN, DS, LT. Data curation: CN, DS, LT, SM, AV. Formal analysis: CN. Software: CN. Investigation: CN, DS, LT. Methodology: CN, LT, DS, CCM. Resources - CCM. Writing – original draft preparation: CN, DS, LT. Writing – review & editing: SM, AV, CCM.

*Competing interests.* The authors have no competing interests to declare

*Acknowledgements.* This study was funded by an ETH Zurich Postdoctoral Fellowship (22-1 FEL-21). CN thanks Alex Lipp for insightful discussions on the Wasserstein distance.





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
