# Peer review of "Controls on zircon age distributions in volcanic, porphyry and plutonic rocks"

_Geochronology, 2024_

## Author Comment (AC1)

**Authors' Response to Review 1 of**

**A statistical analysis of zircon age distributions in volcanic, porphyry and plutonic rocks**

Chetan Nathwani, Dawid Szymanowski, Lorenzo Tavazzani, Sava Markovic, Adrianna L. Virmond, and Cyril Chelle-Michou

*Geochronology Discussions,* `https://doi.org/10.5194/gchron-2024-25`
* * *
**RC:** Reviewers' Comment,     **AR**: Authors' Response

We thank the reviewer for their constructive feedback on our manuscript. The review raises some important omissions from our original manuscript, which did not provide a thorough enough discussion of the uncertainties and assumptions of our modelling approach, which do not consider some aspects such as heat transfer. It also raises the issue of our interpretation of pluton age distributions as akin to 'monotonic cooling' which requires much more in depth discussion. We provide detailed comments to each point raised by the reviewer below.

This paper presents a comprehensive statistical meta-analysis of CA-ID-TIMS zircon age data, identifying differences in the skewness of age distributions across different magmatic environments. The analysis suggests that volcanic and porphyry age distributions often show left-skewed patterns, whereas plutonic distributions tend to be right-skewed. While there is considerable overlap between the age distributions, the overall trend appears consistent and aligns with intuition based on cooling rates. The empirical confirmation of such expectations is valuable.

The paper's strengths lie in its rigorous statistical approach, including quantitative assessments of potential biases and uncertainties. Although not new, the effect of subsampling of zircons is studied, which is useful in a practical sense. Most of the relevant literature is cited, the methods are clearly explained, and figures are well prepared making it straightforward to follow the manuscript.

The authors extend their analysis by proposing an explanation for the observed skewness differences, using a thermodynamic model combined with zircon saturation and bootstrapping techniques. Based on this the authors suggest that eruptive sampling of a monotonously cooling magma body is an important process in controlling natural zircon age distributions, with implications for using these distributions to infer magmatic fluxes from thermal modelling. However, I find this aspect of the study problematic for several reasons:

It is well established based on thermal modelling that simple, monotonous cooling of a magma body will not produce zircon crystallization durations like those shown in your compilation. Most plutons are assembled incrementally, with even small, shallow laccoliths exhibiting zircon age distributions that are inconsistent with simple, homogeneous cooling at the emplacement level. Thus, attributing a significant role to eruptive subsampling in a monotonously cooling magma reservoir has limited relevance for explaining the observed zircon age distributions.

We understand the point of the reviewer here that, according to models, monotonic cooling of small scale plutons might not be able to produce >100 kyr durations of zircon crystallisation. We do not intend to state that large plutons are assembled as one batch of magma emplaced in cold crust (e.g. diapiric emplacement), which is not possible. However, it is a key observation of our study that many plutonic systems exhibit negative skew at a hand-sample scale that is consistent with monotonic cooling (Watson 1996, Keller et al. 2017). There could be a number of factors that reconcile these diverging observations and we have added an

extended discussion to the revised manuscript. Perhaps, (1) the incremental pluton assembly and thermal maturation of the crust required to form a large volume of magma in the upper crust can eventually sustain some degree of monotonic cooling on a hand-sample scale. This would however not be true on a pluton scale where the cumulative zircon distribution across the whole pluton might be dominated by the cooling effect and have young skew. (2) One may also consider that some intrusions considered in this study, such as the Bergell intrusion, were emplaced at mid-crustal depth. At this emplacement level, given the thermal state of the surrounding crust, >100 kyr zircon crystallization might be a plausible occurrence even at the hand-sample scale. (3) We also note that although monotonously cooled bodies are assembled incrementally, the timescale of assembly may be significantly shorter than the timescale of cooling.

This issue is even more problematic when considering that heat transfer modelling shows that monotonous cooling of an intrusion does not necessarily produce a right skewed zircon age distribution in the first place. Using thermodynamic modelling and bootstrapping to obtain synthetic zircon ages, you assume that the thermal evolution can be described by a single time-temperature curve. If the heat equation is used instead to model cooling of an intrusion in 3D, we'll find that the magma spends the longest time at temperatures with smallest temperature gradient to the surroundings. This shifts the synthetic zircon distribution to the left depending on the conditions.

We believe the reviewer here refers to the paper of Schmitt et al. 2023 who show that their thermal models coupled with zircon saturation models produce age distributions with a young skew. As the reviewer points out, this is at odds with the shape of age distribution predicted by simple thermodynamic and zircon solubility models (Watson 1996, Keller et al. 2017). Our study shows that natural zircon age distributions in plutonic rocks tend to display an old skew (Fig. 5), as was previously noted by Samperton et al. (2015) for the Bergell intrusion and later by Tavazzani et al. (2023) for a selected number of plutonic bodies. Assuming that the modeling effort of Schmitt et al. (2023) correctly predicts the coupled thermal-zircon saturation evolution of a cooling intrusion, the reasons for this discrepancy could be due to many different factors:

1. Although a batch of magma might spend the longest time at near-solidus temperatures, the crystallisation of zircon may still proceed according to the predictions of Watson (1996) where most zircon mass precipitates upon zircon saturation

2. In plutons, zircon crystallised closer to the solidus may be hosted in melt pockets rather than armoured by major rock-forming phases. This may lead to preferential "loss" of younger zircons during melt extraction episodes to feed volcanic eruptions or shallower intrusions (e.g. porphyry dikes), preferentially leaving behind the earlier-formed zircon in crystals and thus generating an old skew

3. In complex natural cases, zircons sampled in individual hand samples are likely to reflect the juxtaposition of zircon from various stages of a trans-crustal magmatic system and therefore comparisons with thermal/statistical models of zircon crystallisation in a single magma reservoir cooling in the upper crust are not necessarily valid

4. The overall cooling of an intrusion may not be a monotonic time-temperature curve due to heat transfer to the surrounding crust, but on a hand-sample scale (i.e. the volume of material considered in any geochronology study) it may proceed according to monotonic cooling.

Of course, we cannot provide a sole answer to this issue, but we have enhanced the discussion with the aforementioned points. A few paragraphs and a new section have been added to the revised manuscript.

When it comes to calculating synthetic zircon age distributions, the spatial and temporal dynamics of heat

transfer are hugely important. Given that heat transfer is ignored in your modelling, I don't think that the presented cooling, recharge and subsampling scenarios are particularly insightful.

We emphasise that the underlying zircon crystallisation distributions from which our bootstrap sampling operates cover the extremes of skew that might be possible for zircon crystallisation, with monotonic cooling (extreme old skew; e.g. Fig. 7) and repeated recharge (extreme young skew; e.g. Fig. 9 right panel) scenarios. One of the main conclusions of our paper is that truncation plays a key role in generating different skew of plutonic and volcanic/porphyry age spectra, and that this is not dependent on the mechanism by which the underlying distribution is generated (i.e. heat transfer modelling included or not). Although there may be disagreement on the validity of the "monotonic cooling" distribution, it is consistent with what is observed in plutonic zircon age spectra and is a valid basis from which we can perform bootstrap sampling.

The study's words of caution against using zircon age distributions to infer magma fluxes via thermal modelling in volcanic settings may mislead readers. Specialists in this method may find this feedback irrelevant as heat transfer is not considered, while non-experts might be encouraged to discredit the results of these studies. Rather than discouraging the approach, the study should acknowledge the limitations of its current modelling and emphasize that magma flux quantification requires thermal models accounting for complex magmatic processes and 3D heat transfer.

We agree with the reviewer that we must add a more comprehensive discussion of the limitations of our forward models, and that they do not consider heat transfer. However, we would like to remark that we are not stating that inversion of volcanic zircon age spectra using 3D heat transfer results to obtain magma fluxes is a flawed workflow. Rather, our word of caution is that if the effect of truncation is not considered, the magma flux estimates obtained might be inaccurate. We realised that the previous wording might have been blunt and therefore have been more specific and less critical of these studies in our revised manuscript.

I do not suggest re-running the entire study with thermal modelling but rather recommend a more detailed discussion of the assumptions and limitations inherent in the thermodynamic modelling approach used here. Additionally, based on the comments above, revisiting the conclusions on the relevance of the proposed truncation process and on magma flux quantification, as well as exploring other potential causes for skewness differences—such as emplacement depth—would enhance this paper. For instance, could left-skewed plutonic distributions correspond to shallower intrusions with rapid cooling rates?

We agree that a more detailed discussion of our assumptions and limitations is required. We maintain that truncation does have a large effect but have toned down the criticism of magma flux quantification. As the reviewer recommends, we have also added a more detailed discussion on potential other causes for skewness differences (e.g. melt removal to feed a volcanic eruption, mixing of zircon populations during the eruption, depth of emplacement). We note though that although the depth of emplacement would certainly impact the duration of zircon crystallisation due to slower cooling, we cannot be certain of a large effect on the skew.

I would be happy to support publication of this article if the authors address the issues raised above.

Other comments:

L12-14 At this stage I've got lost without reading the rest of the paper. Could you already here briefly explain how you go from bootstrapping to inferring the processes controlling the difference between volcanic/porphyry zircons on one side and plutonic zircons on the other.

We agree the link between bootstrap sampling (forward modelling) and the natural age distributions is missing here. We have added now a sentence to mention that the synthetic distributions are also cast to PCA space of the dissimilarity matrix of natural data to permit comparison.

L14-15 Here as well, how do you find that higher fluxes can contribute to the skew?

This is based on underlying zircon age distributions generated under different recharge scenarios. We agree it would be useful to add this information to the abstract. Please see the following revised sentences in the abstract:

L34-36 I don't think we can say this with any certainty at this stage. Yes, the early work (Caricchi et al. 2014, 2016) seemed to suggest that there are higher fluxes at work in large eruptive systems compared to plutons, but it turned out that the originally proposed methodology, based on similarity of zircon age distributions (the topic of this work), is too much impacted by statistical biases to reconstruct magma fluxes. That is why other authors used a different sort of approach based the age span, temperature distributions and geological constraints to recover fluxes. A second complication in all this is that different authors use different measures of fluxes such as volume fluxes km3/yr and others area normalized fluxes km3/km2/yr. If we consider Toba as an example (Liu et al. 2021), reconstructed volumetric fluxes are one of the highest ever quantified, while the area normalized flux is less than a typical stratovolcano. This makes a huge difference and is also not considered in the early papers that claim systematically different fluxes for plutons and caldera forming systems.

We agree with the reviewer that this cannot be said with any certainty. The papers of Caricchi et al. (2014, 2016) were a key motivation behind our study and we do feel it is important to mention their initial findings here. We acknowledge that the later papers we cited here (Weber et al. 2020 and Schmitt et al. 2023) use a slightly different approach including additional components such as Ti-in-zircon thermometry.

L47-49 I have a somewhat strong opinion on this but to me this assumption does not make sense at all. Grounding the comparison of zircon age distributions on autocryst-antecryst differences is rather misleading. There is plenty of evidence that magmatic systems are built incrementally by repeated injection of magma batches. Thermally it is quite difficult to produce any long ranging zircon age populations without incremental assembly, given that large single reservoirs would lose heat too quickly. A single magmatic system can therefore contain multiple subsystems that may or may not show some chemical differences in zircon chemistry given variations in recharge magma compositions, differences in partitioning of trace elements in different parts of a reservoir, depth-dependent differences in crustal melting efficiency, and many other factors that would lead to distinct differences in the chemistry of zircons that then could be interpreted as antecrysts. This does not mean that we can't make sense of the age distributions or have to exclude these crystals. None of this is a real problem if the goal is to reconstruct magmatic fluxes or to understand the general dynamics of magmatic systems.

We understand that there is a philosophical issue here in what an antecryst actually represents. As mentioned later on, the exclusion of antecrysts may be detrimental to the understanding of the magmatic system. However, what we mean here is that when zircons are analysed in a rock and compared between different magmatic systems, we need to have a consistent way of comparing these. Some rocks may contain zircon crystallised at multiple depths from multiple pulses of magmatism, which are entrained on ascent, whereas other rocks may contain zircon only crystallised from the latest pulse and do not entrain grains from multiple depths. This may not necessarily reflect changes in magma flux, but could rather be related to what is sampled by the magma. Some antecrysts may crystallise at completely different depths to the main zircon population and would not be sensible to use them in understanding the temporal-thermal evolution of the upper crustal system. Therefore we try to systematically filter the main, youngest population of zircon to isolate zircons from the youngest magmatic pulse (i.e. autocrystic *s.l.*). This strengthens the comparison between age spectra of different magmatic systems. We do however recognize that the original sentence here needs rewording as it was a bit misleading since our method is not the only contribution aimed at comparing zircon age distributions.

L78 Antecrysts cannot be more easily identified with this approach but xenocrysts can.

This is fair enough, at this stage of the manuscript we do not want to enter the antecryst debate so have switched the wording here to give xenocrysts as the example

L84-88 The term "magmatic event" is not right here given that you consider the entire evolution or at least large parts of the history of a magmatic system e.g. over 500 ka in the example you give above.

Thanks for pointing this out, we agree it is not a useful term as a magmatic event could even refer to a volcanic eruption. The paper of Miller et al. (2007) which defined this term uses 'pulse'. We have therefore used this term instead:

L85-89 See discussion about antecrysts above.

See our comment above.

L92 "making the autocryst–antecryst divide ambiguous, and possibly detrimental to the understanding of the underlying system". Okay, couldn't have said it better.

Thanks.

L106 "... ECDF older than the two youngest dates (to ignore age gaps at the young end of the distribution which are not considered here):". That is fine but I wonder why only the two youngest dates? Don't you have any gaps that could be beyond that threshold in the young part of the distribution?

We agree that the setting of this parameter is a bit empirical for now. Because we deal with young datasets, and any obvious Pb loss was removed manually, we in general did not have to deal with young anomalies during the filtering process. We set this value to 2 in order to ignore any anomalously young dates that may persist, but also did not want to use a higher value as this then increases the weighting of old outliers in the distribution and filters these overly rigorously. Setting this number more objectively is undebatably challenging.

L150-153 A minimum of 10 zircons to resolve the shape of the distribution seems rather optimistic given the bootstrapping you show later in Fig. 7.

Indeed we agree that what is shown in Figure 7 is not hugely convincing for the choice of 10 zircons. However, as we discuss in the manuscript and as shown in Figure S8, this choice of minimum zircons can still capture the underlying distribution, even though there is considerable scatter that we quantify. Even with this threshold, we can still see broad differences between different rock types. As we mention in the manuscript, it would be ideal to set the threshold higher but we would lose most of our data and this study would not be possible. As more datasets are published and compiled we hope the filtering process can become even more rigorous and will allow us to further refine the differences that exist between different magmatic products.

L253-254 "overall divide" is somewhat of an overstatement given that most of the volcanic and plutonic distributions overlap. I agree that there is a tendency for the volcanics and porphyry to be left skewed and right skewed for the plutonics.

This is a fair point. We have adjusted this sentence in the revised manuscript to reduce the emphasis on this separation being very clear.

L359-374 1) Given that there are only a few, if these are the part of second older age population could be tested by simply looking at the relevant datasets.

This is a fair point, though it is difficult to say for sure just from looking at the spectrum whether it comes from a second older population or is just from sampling statistics. We have therefore added a sentence to state that.

2) Your bootstrapping plot (Fig. 7) shows quite some variability also for the n=30 case. My suspicion is that you could also explain the extreme cases if you slightly change the shape of the theoretical sampling distribution and/or just use more iterations.

True. But this is our objective with sampling different underlying age distributions generated with repeated recharges (Fig. 9). Even with 100 iterations and a very young skewed distribution it is still difficult to generate those distributions (Fig. 9 right panel). So we maintain our idea that incorporating an older population could be an important factor. As the reviewer mentions earlier, these antecrysts are probably quite common and difficult to resolve.

L338-444 FYI. The effect of the specific zircon saturation curve in use is quite large when calculating synthetic zircon distributions based on thermodynamic modelling. However, when using heat transfer the effect of the temperature distribution through time is much more important. In this case, switching the zircon saturation model around has only a minor effect on what the synthetic zircon distribution looks like, so this is secondary at best.

We agree with the reviewer here. This section does not discuss different zircon saturation curves and instead deals with the effect of greater numbers of recharge in a magma reservoir and shifting the peak of zircon crystallisation towards the solidus. Our point is to show what the shift is in the modelled age distribution with increasing number of recharges.

References:

Liu, P. P., Caricchi, L., Chung, S. L., Li, X. H., Li, Q. L., Zhou, M. F., ... & Simpson, G. (2021). Growth and thermal maturation of the Toba magma reservoir

References:

Caricchi, L., Simpson, G. and Schaltegger, U., 2014. Zircons reveal magma fluxes in the Earth's crust. Nature, 511(7510), pp.457-461.

Caricchi, L., Simpson, G. and Schaltegger, U., 2016. Estimates of volume and magma input in crustal magmatic systems from zircon geochronology: the effect of modeling assumptions and system variables. Frontiers in Earth Science, 4, p.48.

Keller, C.B., Schoene, B. and Samperton, K.M., 2018. A stochastic sampling approach to zircon eruption age interpretation. Geochemical Perspectives Letters (Online), 8(LLNL-JRNL-738859).

Miller, J.S., Matzel, J.E., Miller, C.F., Burgess, S.D. and Miller, R.B., 2007. Zircon growth and recycling during the assembly of large, composite arc plutons. Journal of Volcanology and Geothermal Research, 167(1-4), pp.282-299.

Schmitt, A.K., Sliwinski, J., Caricchi, L., Bachmann, O., Riel, N., Kaus, B.J.P., Cisneros de Léon, A., Cornet, J., Friedrichs, B., Lovera, O. and Sheldrake, T., 2023. Zircon age spectra to quantify magma evolution. Geosphere, 19(4), pp.1006-1031.

Samperton, K.M., Schoene, B., Cottle, J.M., Keller, C.B., Crowley, J.L. and Schmitz, M.D., 2015. Magma emplacement, differentiation and cooling in the middle crust: Integrated zircon geochronological–geochemical constraints from the Bergell Intrusion, Central Alps. Chemical Geology, 417, pp.322-340.

Tavazzani, L., Wotzlaw, J.F., Economos, R., Sinigoi, S., Demarchi, G., Szymanowski, D., Laurent, O., Bachmann, O. and Chelle-Michou, C., 2023. High-precision zircon age spectra record the dynamics and evolution of large open-system silicic magma reservoirs. Earth and Planetary Science Letters, 623, p.118432.

Watson, E.B., 1996. Dissolution, growth and survival of zircons during crustal fusion: kinetic principals, geological models and implications for isotopic inheritance. Earth and Environmental Science Transactions of the Royal Society of Edinburgh, 87(1-2), pp.43-56.

Weber, G., Caricchi, L., Arce, J.L. and Schmitt, A.K., 2020. Determining the current size and state of subvolcanic magma reservoirs. Nature Communications, 11(1), p.5477.

---

## Author Comment (AC2)

**Authors' Response to Reviewer 2 of**

**A statistical analysis of zircon age distributions in volcanic, porphyry and plutonic rocks**

Chetan Nathwani, Dawid Szymanowski, Lorenzo Tavazzani, Sava Markovic, Adrianna L. Virmond, and Cyril Chelle-Michou

Geochronology Discussions, https://doi.org/10.5194/gchron-2024-25

RC: Reviewers' Comment, AR: Authors' Response

We thank the reviewer for their supportive and constructive feedback on our manuscript and we are pleased they find it to be useful contribution to the geochronology community. They provide a series of very useful minor comments on the manuscript which we respond to individually below.

**General Comments:**

This manuscript addresses a critical issue in modern igneous petrology: How best to understand the measured timescales of crystallization in igneous rocks. The lens here is an important subset of this problem, zircon crystallization timescales, when measured using whole grain analyses at precisions that are much higher than the crystallization timescales.

In this work, the authors produce a large number of carefully curated U-Pb datasets, present a quantitative model for comparing their distributions (Wasserstein difference) and reducing the degrees of freedom in them (PCA), extract trends in the data that correlate broadly with the type of igneous system, and then compare these trends with predictions they make based on simple forward models. I found it to be a very clearly written paper, with good explanations of the quantitative techniques and explanations of how they related to the underlying datasets. The figures are high quality, the writing is excellent, and the choice of topic is well within the remit of Geochronology.

**Specific Comments:**

Like I said above, this is a very good paper. The only real issue is that the authors don't address that there are important differences between their forward models and real processes that drive crystallization and preservation of crystallization age distributions. As I'm writing this I note that this is something that the first reviewer also pointed out, so I won't spend a lot of time on it, but I think this is probably worth addressing qualitatively, for completeness. Both in terms of what the T-t history is like, and additional complications such as those highlighted by Klein and Eddy, and biases in the measured zircon record. I appreciate that it may not be possible to include those complications at this time in a quantitative model, but they are both real and important.

We have incorporated more of a discussion on the differences between our modelling and real processes, including much more detail on what could explain the difference between the results of our statistical analysis and thermal models from other studies. A more detailed response can be found in our reply to Reviewer 1. We have also made the reader aware of the caveat of our study that it does not consider some of the complications highlighted by Klein and Eddy (see below).

**Specific comments/technical corrections**

L43-44: It's worth noting that sampling biases may also play a role. I can't speak for every analyst, of course,

but typically the largest and highest quality crystals are selected from a population. The size bias alone may be significant, as is the fact that whole grains modified by CA bias U-Pb dates by U concentration, dissolve components that are accessible to HF during the partial dissolution phase of CA, and integrate the remaining grain.

We agree that it would be useful to make a note of some additional biases. We have thought not to get into too much detail about human biases invoked by ID-TIMS as this is something that will not be unique to a specific type of sample, or the chemical abrasion biases as we do not want to convolute and confuse the paper. We do think however that it is worth noting the fact that there is a bias of bulk grain analyses to younger ages due to the volumetric differences in core to rim, as pointed out by Klein and Eddy (2023. We specify therefore that our model assumes instantaneous crystallisation of zircon

L120: This is an excellent summary of the method.

**Thank you.**

L134-137: I think it's fair to stick to one type of dataset but the objections to 230/238 geochronology aren't strong. Variable and asymmetric uncertainties aren't difficult to deal with, and initial [230/238] are determined in many high quality datasets (and doesn't vary infinitely). There are good reasons that plutonic datasets are rare (or absent?) and that's because the technique is exclusively applicable to young rocks, and that's a reason enough alone to exclude it.

This is fair. By far the most important reason to not include these data is their being limited to volcanic samples (simply because there are few if any plutons this young and already exposed). The asymmetric uncertainties are not a big deal, but the lack of a robust way to estimate melt-(230/238) corresponding to each zircon date remains a major source of uncertainty. We have adjusted this part to reflect this in the revised manuscript.

L159: This is very good.

**Thanks!**

L201-206: I'm a bit confused by this paragraph and unsure as to what I'm supposed to take away from it as a reader. It describes a technique that is not used, justifies why it is suboptimal, but then in the last sentence tells me that the results are the same? I have no specific objection here but I think the authors might want to clarify for a reader what they should take away from this. For me personally, as a geochemist who has used MDS in the past, I'd be just as happy if this paragraph didn't exist? I don't think many readers will look at this and wonder why they didn't use MDS.

This is a fair point that it does not necessarily add anything to the manuscript and the reader will not wonder why the readers did not use MDS. We therefore have accepted the reviewer's suggestion and removed this from the revised manuscript.

L220-221: Minor point but mixing and matching confidence/coverage intervals is confusing to a reader (the parameter fit uncertainties are at a coverage factor of 1 and the result is at 2).

**Yes, this is a valid point thanks, the standard errors on the fit parameters have now been updated to 2SE in line with the $2\sigma$ of the output.**

L221: This is a really useful result, and I could see it being something used by others in future work – it might be worth adding a bit more context for those who do? Can you specify the time range that this is applicable to, and if the fit parameters uncertainties are correlated? In the introduction you list 0.1-4567 Ma, but that's

probably not relevant here (if I do the calculation right it's 40% at 0.1 Ma.)

This is calculated over the range of dates included in the compilation we used (Markovic et al.) but only up to 1000 Ma (to keep it relevant to the maximum important for our study and to avoid the fact that older dates would use Pb/Pb). We have specified this now in the revised manuscript. We agree it can be useful if one is interested in the general development of uncertainties with age, but we would not take it too far. Specifically, this shows the mid-range uncertainties in dates from multiple labs, and it uses a relatively simple fit. So real uncertainties available to any given lab may be much better or much worse.

It's also worth noting that these uncertainties differ markedly from those listed rather optimistically on line 138 (e.g., if I did the calculation correctly, it bottoms out at 0.11% in the Neoproterozoic). I'd recommend revisiting line 138 to bring it in line with the actual uncertainties parameterized in this equation – that's more of a boilerplate ID-TIMS boast than a realistic assessment of actual typical precisions from young zircon populations, where for even low-blank labs, the precision ends up being limited by the blank isotopic composition variability.

While the value in line 138 is very much "a boilerplate ID-TIMS boast", a realistic assessment of typical precisions based on that same compilation shows that it's actually fairly close to truth for all but the very youngest samples - see Fig. 11e in Markovic et al. We will adjust line 138 to reflect that reality better.

L235-240: It's worth revisiting the issues raised by Klein and Eddy at this point. Not that they need to be incorporated but just as a caveat for the reader.

Yes agreed that it would be useful to mention their work. We have decided it would best to bring this into the Methods, where we state that our model, for simplicity, assumes the instantaneous growth of each zircon and ignores the protracted growth of each crystal and the inherent bias of bulk grain analysis to younger ages. We also add a reference to Melnik and Bindeman (2018) which discusses this.

L268: Please be cautious here in assuming that natural processes won't result in gaussian distributions. The central limit theorem applies to natural processes as well – the samples we recover average a range of disparate pieces of data and processes, and averaged samples from any distribution are gaussian.

Very fair point. This requires some careful rewording. What we mean here is that our threshold for  $\Delta T/\sigma$  is probably valid as the lowest  $\Delta T/\sigma$  in our filtered datasets still exhibit skew indicating they are not dominated by gaussian uncertainty (and would thus typically show normal distributions). We have rephrased to remove the statement of non-normal distributions being dominated by geological dispersion.

L305: Can you compare this to other estimates if you are aware of any? (This isn't a leading question, I'm can't find one myself)

There are some estimates made by Caricchi et al. (2016) who compared the mean, mode and median of a zircon age distribution with different numbers of zircon sampled. When sampling a normal distribution, they show that already with 10 zircons sampled, the mean, mode and median are similar to the underlying distribution. For a skewed distribution (which is what our sampling is performed on), they show that significant improvement is made when up to 30 zircons are sampled. Their sampling is performed at 100 kyr and we are unclear how/if uncertainties are propagated onto these results. Our study of the effect of number of zircons sampled is based on comparing the entire shapes of distributions rather than mean, mode and median. Given the difference in approaches and different ages under which the modelling is performed, we decided not to make a comparison in the text here as getting into the details will distract a bit the reader from the key point.

L412: This seems like an unnecessary swipe at LA-ICPMS labs. Using data that is accurate (including

accurate uncertainties) is obviously critical and there's no reason to cast aspersions on a specific technique. Peak hopping TIMS Pb data on Phoenix instruments have uncertainties on isotope ratios underestimated by 1.7x due to autocorrelation (and some other factor on any other TIMS), but that fact is equally out of place here. LA-ICPMS is an easy target because of the large number of practitioners and therefore the larger n of outliers.

We do not believe we are making an unwarranted swipe at the LA-ICP-MS technique but we think it is very important to note here that using spot analyses to reconstruct an age distribution in these young rocks may be more challenging than one might think from just looking at the reported uncertainties. Quite a lot of studies report single-spot  $2\sigma$  uncertainties at about 0.5% (which is likely just considering analytical uncertainty), but when taking into account repeatability of reference material analyses (which is a good approximation for systematic uncertainties associated with 'matrix effects' in LA-ICP-MS analyses) this is actually within the range of 2-4% which is quite a substantial difference (Sliwinski et al., 2022). Hence, we feel it appropriate here to mention as we want to avoid people overinterpreting age distributions from LA-ICP-MS data with under-estimated uncertainties. We have replaced the word "neglected" with "do not account for" in order to soften these words for the LA-ICP-MS community.

The issue of potentially underestimated uncertainties in peak-hopping TIMS Pb analyses is not insignificant as it will affect our threshold  $\Delta t/\sigma$  by reducing the resolution of some datasets. It is however hard to address for now since it is not documented to a point where we could, for example, apply a correction factor to each dataset. We hope progress in data reduction algorithms for ion counter analyses will soon catch up with the needs of the community in that regard.

---

## Author Response (AR2)

Dear Dr. Nathwani

thank you resubmitting a carefully revised version of your manuscript. I really appreciated that you submitted a clear, thorough, well-illustrated and well-argued paper in the first place.

The modifications based on the review comments further improved the paper and it is now accepted for publication.

I chose "Publish subject to technical corrections" to give you the opportunity to make the modifications suggested by the Associate Editor. The changes are at your discretion.

In the versions your submitted (clean, and the one with changes visible), Figure 3 is missing. It is possible that there is a problem with my computer and it cannot upload/display the figure. Make sure the figure is included in your final printed version of your paper.

With kind regards
Klaus Mezger

Dear Prof. Mezger,

Thank you for the handling of our manuscript and for your positive feedback. We have made a change in response to the associate editor (see below also). We cannot find an issue with Figure 3 on our computers on the pdf we uploaded. In any case, the figures are now uploaded separately at this stage so it will be fine.

Best wishes,

Chetan Nathwani

Brenhin Keller

This revised manuscript satisfactorily addresses my and the reviewers comments in my estimation

Thanks for adding the sub-grain fracturing discussion and superscripts in the table. I might quibble a bit with the phrasing that sub-grain fracturing will "skew the age distribution to older ages" -- it may well result in a distribution with a greater skew towards older ages in the statistical sense, but "skewed" in vernacular terms implies this to be an error, but if anything the fractured distribution will be more representative of the true zircon crystallization distribution than an equivalent whole-grain age distribution. At the same time, by mass conservation, if some fragments are older cores, others will have to be younger rims, so more generically fracturing will also increase the apparent age dispersion by resolving intra-grain heterogeneity. Personally I would flesh this discussion out just a bit more still, but I'll leave this to your taste -- "technical corrections" means no further review by me.

A fair point, and agreed better to add some more detail here. We have revised as follows:

An additional factor is that a small number of datasets (Barboni et al., 2015; Samperton et al., 2015) contain fractured sub-grain analyses which **can resolve intra-grain heterogeneity and thus better represent the true zircon crystallisation distribution** (Klein and Eddy, 2023). For example, Samperton et al. (2015) present fractured sub-grain analyses where inner fragments show considerably older ages than external fragments (up to 500 kyr) which may explain the remarkable consistency of their age distributions with that predicted from zircon solubility and thermodynamics (Keller et al., 2018). However, old skew is observed in plutonic age distributions where sub-grain fragments were not analysed (e.g., Barboni et al., 2015; Kinney et al., 2021) suggesting it may not be a primary factor.

Considering the manuscript, reviews, and responses to reviews, I believe a version of the manuscript that has been revised in line with the author responses to reviews would represent a valuable contribution suitable for publication in GChron.

Beyond the issues discussed in the reviews and responses thereto, I would just like to highlight one other factor which hasn't been much discussed here yet, regarding the role of whole-grain vs sub-grain TIMS analyses. So far this has been alluded to in the context of possible volumetric bias towards grain rims, but less so in the context of the more fundamental issue of just how much time is being averaged in a single date. Fortunately however, some TIMS datasets reduce the time-integration of a given analysis by microfracturing zircons into sub-grain fragments. In particular, I have long wondered if the unusually good agreement of the Samperton 2015/2017 dataset with the "idealized" saturation model might have anything to do with the extensive sub-grain micofracturing in that dataset, but haven't run into enough other high-resolution subgrain TIMS datasets for much of a a systematic comparison.

I think my biggest request here would be to add a column to Table 2 denoting which of these TIMS datasets are predominantly whole-grain datasets and which are predominantly microfractured datasets, and perhaps some minor discussion of what effect if any this might have on the resulting distributions (presumably, some sort of convolution in statistical terms?).

Compared to this, I'm less worried about the issue of volumetric bias towards grain rims as alluded to in the context of Klein and Eddy -- i.e., yes, that would certainly be an issue with respect to age distribution bias if zircons grew at a constant radial growth rate, but it's not apparent that there is any physical process basis on which to expect constant radial zircon growth; diffusion-limitation would lead to something more like constant volumetric (not radial) growth rate, and saturation-limitation would lead to volumetric growth rates that match a saturation curve.

One other minor request for Table 2 (and associated discussion: the "Mean $\Delta t$" and "Mean $\Delta t/\sigma$" should probably be renamed to "Mean Apparent $\Delta t$" and "Mean Apparent $\Delta t/\sigma$", assuming those are derived by simply subtracting youngest from oldest dated zircon; perhaps a fine point but I think it's important to differentiate between the idealized/mathematical "$\Delta t$" which equals the true duration of zircon crystallization but is very hard to truly know, and the "apparent $\Delta t$", i.e. oldest-youngest zircon, which is an imperfect estimate of the former. For durations that have been estimated by some (e.g.) Bayesian or other statistical approach, then I would suggest something along the lines of "Mean estimated $\Delta t$" as the analogous preferred term.

Thanks for the constructive feedback. Only the studies of Samperton et al. (2017) and Barboni et al. (2015) report sub-grain analysis to our knowledge. We have marked this now in the Table. We have added a sentence in the discussion to mention this but we do not think it will be a major factor, since whole-grain zircon analyses can also exhibit old skew. The Barboni et al. paper also just analyses rims, not cores, so this would bias towards younger ages.

We updated the mean "apparent" $\Delta t$ and $\Delta t/\sigma$ in the text and table as requested.

The discussion of the bias towards younger ages due to volumetric effects is only described in the case of us assuming each zircon grain crystallizes instantaneously in our models.

**Authors' Response to Review 1 of**

**A statistical analysis of zircon age distributions in volcanic, porphyry and plutonic rocks**

Chetan Nathwani, Dawid Szymanowski, Lorenzo Tavazzani, Sava Markovic, Adrianna L. Virmond, and Cyril Chelle-Michou

*Geochronology Discussions,* `https://doi.org/10.5194/gchron-2024-25`

**RC:** Reviewers' Comment,    **AR**: Authors' Response

We thank the reviewer for their constructive feedback on our manuscript. The review raises some important omissions from our original manuscript, which did not provide a thorough enough discussion of the uncertainties and assumptions of our modelling approach, which do not consider some aspects such as heat transfer. It also raises the issue of our interpretation of pluton age distributions as akin to 'monotonic cooling' which requires much more in depth discussion. We provide detailed comments to each point raised by the reviewer below.

This paper presents a comprehensive statistical meta-analysis of CA-ID-TIMS zircon age data, identifying differences in the skewness of age distributions across different magmatic environments. The analysis suggests that volcanic and porphyry age distributions often show left-skewed patterns, whereas plutonic distributions tend to be right-skewed. While there is considerable overlap between the age distributions, the overall trend appears consistent and aligns with intuition based on cooling rates. The empirical confirmation of such expectations is valuable.

The paper's strengths lie in its rigorous statistical approach, including quantitative assessments of potential biases and uncertainties. Although not new, the effect of subsampling of zircons is studied, which is useful in a practical sense. Most of the relevant literature is cited, the methods are clearly explained, and figures are well prepared making it straightforward to follow the manuscript.

The authors extend their analysis by proposing an explanation for the observed skewness differences, using a thermodynamic model combined with zircon saturation and bootstrapping techniques. Based on this the authors suggest that eruptive sampling of a monotonously cooling magma body is an important process in controlling natural zircon age distributions, with implications for using these distributions to infer magmatic fluxes from thermal modelling. However, I find this aspect of the study problematic for several reasons:

It is well established based on thermal modelling that simple, monotonous cooling of a magma body will not produce zircon crystallization durations like those shown in your compilation. Most plutons are assembled incrementally, with even small, shallow laccoliths exhibiting zircon age distributions that are inconsistent with simple, homogeneous cooling at the emplacement level. Thus, attributing a significant role to eruptive subsampling in a monotonously cooling magma reservoir has limited relevance for explaining the observed zircon age distributions.

We understand the point of the reviewer here that, according to models, monotonic cooling of small scale plutons might not be able to produce >100 kyr durations of zircon crystallisation. We do not intend to state that large plutons are assembled as one batch of magma emplaced in cold crust (e.g. diapiric emplacement), which is not possible. However, it is a key observation of our study that many plutonic systems exhibit negative skew at a hand-sample scale that is consistent with monotonic cooling (Watson 1996, Keller et al. 2017). There could be a number of factors that reconcile these diverging observations and we have added an

extended discussion to the revised manuscript. Perhaps, (1) the incremental pluton assembly and thermal maturation of the crust required to form a large volume of magma in the upper crust can eventually sustain some degree of monotonic cooling on a hand-sample scale. This would however not be true on a pluton scale where the cumulative zircon distribution across the whole pluton might be dominated by the cooling effect and have young skew. (2) One may also consider that some intrusions considered in this study, such as the Bergell intrusion, were emplaced at mid-crustal depth. At this emplacement level, given the thermal state of the surrounding crust, >100 kyr zircon crystallization might be a plausible occurrence even at the hand-sample scale. (3) We also note that although monotonously cooled bodies are assembled incrementally, the timescale of assembly may be significantly shorter than the timescale of cooling.

This issue is even more problematic when considering that heat transfer modelling shows that monotonous cooling of an intrusion does not necessarily produce a right skewed zircon age distribution in the first place. Using thermodynamic modelling and bootstrapping to obtain synthetic zircon ages, you assume that the thermal evolution can be described by a single time-temperature curve. If the heat equation is used instead to model cooling of an intrusion in 3D, we'll find that the magma spends the longest time at temperatures with smallest temperature gradient to the surroundings. This shifts the synthetic zircon distribution to the left depending on the conditions.

We believe the reviewer here refers to the paper of Schmitt et al. 2023 who show that their thermal models coupled with zircon saturation models produce age distributions with a young skew. As the reviewer points out, this is at odds with the shape of age distribution predicted by simple thermodynamic and zircon solubility models (Watson 1996, Keller et al. 2017). Our study shows that natural zircon age distributions in plutonic rocks tend to display an old skew (Fig. 5), as was previously noted by Samperton et al. (2015) for the Bergell intrusion and later by Tavazzani et al. (2023) for a selected number of plutonic bodies. Assuming that the modeling effort of Schmitt et al. (2023) correctly predicts the coupled thermal-zircon saturation evolution of a cooling intrusion, the reasons for this discrepancy could be due to many different factors:

1. Although a batch of magma might spend the longest time at near-solidus temperatures, the crystallisation of zircon may still proceed according to the predictions of Watson (1996) where most zircon mass precipitates upon zircon saturation

2. In plutons, zircon crystallised closer to the solidus may be hosted in melt pockets rather than armoured by major rock-forming phases. This may lead to preferential "loss" of younger zircons during melt extraction episodes to feed volcanic eruptions or shallower intrusions (e.g. porphyry dikes), preferentially leaving behind the earlier-formed zircon in crystals and thus generating an old skew

3. In complex natural cases, zircons sampled in individual hand samples are likely to reflect the juxtaposition of zircon from various stages of a trans-crustal magmatic system and therefore comparisons with thermal/statistical models of zircon crystallisation in a single magma reservoir cooling in the upper crust are not necessarily valid

4. The overall cooling of an intrusion may not be a monotonic time-temperature curve due to heat transfer to the surrounding crust, but on a hand-sample scale (i.e. the volume of material considered in any geochronology study) it may proceed according to monotonic cooling.

Of course, we cannot provide a sole answer to this issue, but we have enhanced the discussion with the aforementioned points. A few paragraphs and a new section have been added to the revised manuscript.

When it comes to calculating synthetic zircon age distributions, the spatial and temporal dynamics of heat

transfer are hugely important. Given that heat transfer is ignored in your modelling, I don't think that the presented cooling, recharge and subsampling scenarios are particularly insightful.

We emphasise that the underlying zircon crystallisation distributions from which our bootstrap sampling operates cover the extremes of skew that might be possible for zircon crystallisation, with monotonic cooling (extreme old skew; e.g. Fig. 7) and repeated recharge (extreme young skew; e.g. Fig. 9 right panel) scenarios. One of the main conclusions of our paper is that truncation plays a key role in generating different skew of plutonic and volcanic/porphyry age spectra, and that this is not dependent on the mechanism by which the underlying distribution is generated (i.e. heat transfer modelling included or not). Although there may be disagreement on the validity of the "monotonic cooling" distribution, it is consistent with what is observed in plutonic zircon age spectra and is a valid basis from which we can perform bootstrap sampling.

The study's words of caution against using zircon age distributions to infer magma fluxes via thermal modelling in volcanic settings may mislead readers. Specialists in this method may find this feedback irrelevant as heat transfer is not considered, while non-experts might be encouraged to discredit the results of these studies. Rather than discouraging the approach, the study should acknowledge the limitations of its current modelling and emphasize that magma flux quantification requires thermal models accounting for complex magmatic processes and 3D heat transfer.

We agree with the reviewer that we must add a more comprehensive discussion of the limitations of our forward models, and that they do not consider heat transfer. However, we would like to remark that we are not stating that inversion of volcanic zircon age spectra using 3D heat transfer results to obtain magma fluxes is a flawed workflow. Rather, our word of caution is that if the effect of truncation is not considered, the magma flux estimates obtained might be inaccurate. We realised that the previous wording might have been blunt and therefore have been more specific and less critical of these studies in our revised manuscript.

I do not suggest re-running the entire study with thermal modelling but rather recommend a more detailed discussion of the assumptions and limitations inherent in the thermodynamic modelling approach used here. Additionally, based on the comments above, revisiting the conclusions on the relevance of the proposed truncation process and on magma flux quantification, as well as exploring other potential causes for skewness differences—such as emplacement depth—would enhance this paper. For instance, could left-skewed plutonic distributions correspond to shallower intrusions with rapid cooling rates?

We agree that a more detailed discussion of our assumptions and limitations is required. We maintain that truncation does have a large effect but have toned down the criticism of magma flux quantification. As the reviewer recommends, we have also added a more detailed discussion on potential other causes for skewness differences (e.g. melt removal to feed a volcanic eruption, mixing of zircon populations during the eruption, depth of emplacement). We note though that although the depth of emplacement would certainly impact the duration of zircon crystallisation due to slower cooling, we cannot be certain of a large effect on the skew.

I would be happy to support publication of this article if the authors address the issues raised above.

Other comments:

L12-14 At this stage I've got lost without reading the rest of the paper. Could you already here briefly explain how you go from bootstrapping to inferring the processes controlling the difference between volcanic/porphyry zircons on one side and plutonic zircons on the other.

We agree the link between bootstrap sampling (forward modelling) and the natural age distributions is missing here. We have added now a sentence to mention that the synthetic distributions are also cast to PCA space of the dissimilarity matrix of natural data to permit comparison.

L14-15 Here as well, how do you find that higher fluxes can contribute to the skew?

This is based on underlying zircon age distributions generated under different recharge scenarios. We agree it would be useful to add this information to the abstract. Please see the following revised sentences in the abstract:

L34-36 I don't think we can say this with any certainty at this stage. Yes, the early work (Caricchi et al. 2014, 2016) seemed to suggest that there are higher fluxes at work in large eruptive systems compared to plutons, but it turned out that the originally proposed methodology, based on similarity of zircon age distributions (the topic of this work), is too much impacted by statistical biases to reconstruct magma fluxes. That is why other authors used a different sort of approach based the age span, temperature distributions and geological constraints to recover fluxes. A second complication in all this is that different authors use different measures of fluxes such as volume fluxes km3/yr and others area normalized fluxes km3/km2/yr. If we consider Toba as an example (Liu et al. 2021), reconstructed volumetric fluxes are one of the highest ever quantified, while the area normalized flux is less than a typical stratovolcano. This makes a huge difference and is also not considered in the early papers that claim systematically different fluxes for plutons and caldera forming systems.

We agree with the reviewer that this cannot be said with any certainty. The papers of Caricchi et al. (2014, 2016) were a key motivation behind our study and we do feel it is important to mention their initial findings here. We acknowledge that the later papers we cited here (Weber et al. 2020 and Schmitt et al. 2023) use a slightly different approach including additional components such as Ti-in-zircon thermometry.

L47-49 I have a somewhat strong opinion on this but to me this assumption does not make sense at all. Grounding the comparison of zircon age distributions on autocryst-antecryst differences is rather misleading. There is plenty of evidence that magmatic systems are built incrementally by repeated injection of magma batches. Thermally it is quite difficult to produce any long ranging zircon age populations without incremental assembly, given that large single reservoirs would lose heat too quickly. A single magmatic system can therefore contain multiple subsystems that may or may not show some chemical differences in zircon chemistry given variations in recharge magma compositions, differences in partitioning of trace elements in different parts of a reservoir, depth-dependent differences in crustal melting efficiency, and many other factors that would lead to distinct differences in the chemistry of zircons that then could be interpreted as antecrysts. This does not mean that we can't make sense of the age distributions or have to exclude these crystals. None of this is a real problem if the goal is to reconstruct magmatic fluxes or to understand the general dynamics of magmatic systems.

We understand that there is a philosophical issue here in what an antecryst actually represents. As mentioned later on, the exclusion of antecrysts may be detrimental to the understanding of the magmatic system. However, what we mean here is that when zircons are analysed in a rock and compared between different magmatic systems, we need to have a consistent way of comparing these. Some rocks may contain zircon crystallised at multiple depths from multiple pulses of magmatism, which are entrained on ascent, whereas other rocks may contain zircon only crystallised from the latest pulse and do not entrain grains from multiple depths. This may not necessarily reflect changes in magma flux, but could rather be related to what is sampled by the magma. Some antecrysts may crystallise at completely different depths to the main zircon population and would not be sensible to use them in understanding the temporal-thermal evolution of the upper crustal system. Therefore we try to systematically filter the main, youngest population of zircon to isolate zircons from the youngest magmatic pulse (i.e. autocrystic *s.l.*). This strengthens the comparison between age spectra of different magmatic systems. We do however recognize that the original sentence here needs rewording as it was a bit misleading since our method is not the only contribution aimed at comparing zircon age distributions.

L78 Antecrysts cannot be more easily identified with this approach but xenocrysts can.

This is fair enough, at this stage of the manuscript we do not want to enter the antecryst debate so have switched the wording here to give xenocrysts as the example

L84-88 The term "magmatic event" is not right here given that you consider the entire evolution or at least large parts of the history of a magmatic system e.g. over 500 ka in the example you give above.

Thanks for pointing this out, we agree it is not a useful term as a magmatic event could even refer to a volcanic eruption. The paper of Miller et al. (2007) which defined this term uses 'pulse'. We have therefore used this term instead:

L85-89 See discussion about antecrysts above.

See our comment above.

L92 "making the autocryst–antecryst divide ambiguous, and possibly detrimental to the understanding of the underlying system". Okay, couldn't have said it better.

Thanks.

L106 "... ECDF older than the two youngest dates (to ignore age gaps at the young end of the distribution which are not considered here):". That is fine but I wonder why only the two youngest dates? Don't you have any gaps that could be beyond that threshold in the young part of the distribution?

We agree that the setting of this parameter is a bit empirical for now. Because we deal with young datasets, and any obvious Pb loss was removed manually, we in general did not have to deal with young anomalies during the filtering process. We set this value to 2 in order to ignore any anomalously young dates that may persist, but also did not want to use a higher value as this then increases the weighting of old outliers in the distribution and filters these overly rigorously. Setting this number more objectively is undebatably challenging.

L150-153 A minimum of 10 zircons to resolve the shape of the distribution seems rather optimistic given the bootstrapping you show later in Fig. 7.

Indeed we agree that what is shown in Figure 7 is not hugely convincing for the choice of 10 zircons. However, as we discuss in the manuscript and as shown in Figure S8, this choice of minimum zircons can still capture the underlying distribution, even though there is considerable scatter that we quantify. Even with this threshold, we can still see broad differences between different rock types. As we mention in the manuscript, it would be ideal to set the threshold higher but we would lose most of our data and this study would not be possible. As more datasets are published and compiled we hope the filtering process can become even more rigorous and will allow us to further refine the differences that exist between different magmatic products.

L253-254 "overall divide" is somewhat of an overstatement given that most of the volcanic and plutonic distributions overlap. I agree that there is a tendency for the volcanics and porphyry to be left skewed and right skewed for the plutonics.

This is a fair point. We have adjusted this sentence in the revised manuscript to reduce the emphasis on this separation being very clear.

L359-374 1) Given that there are only a few, if these are the part of second older age population could be tested by simply looking at the relevant datasets.

This is a fair point, though it is difficult to say for sure just from looking at the spectrum whether it comes from a second older population or is just from sampling statistics. We have therefore added a sentence to state that.

2) Your bootstrapping plot (Fig. 7) shows quite some variability also for the n=30 case. My suspicion is that you could also explain the extreme cases if you slightly change the shape of the theoretical sampling distribution and/or just use more iterations.

True. But this is our objective with sampling different underlying age distributions generated with repeated recharges (Fig. 9). Even with 100 iterations and a very young skewed distribution it is still difficult to generate those distributions (Fig. 9 right panel). So we maintain our idea that incorporating an older population could be an important factor. As the reviewer mentions earlier, these antecrysts are probably quite common and difficult to resolve.

L338-444 FYI. The effect of the specific zircon saturation curve in use is quite large when calculating synthetic zircon distributions based on thermodynamic modelling. However, when using heat transfer the effect of the temperature distribution through time is much more important. In this case, switching the zircon saturation model around has only a minor effect on what the synthetic zircon distribution looks like, so this is secondary at best.

We agree with the reviewer here. This section does not discuss different zircon saturation curves and instead deals with the effect of greater numbers of recharge in a magma reservoir and shifting the peak of zircon crystallisation towards the solidus. Our point is to show what the shift is in the modelled age distribution with increasing number of recharges.

References:

Liu, P. P., Caricchi, L., Chung, S. L., Li, X. H., Li, Q. L., Zhou, M. F., ... & Simpson, G. (2021). Growth and thermal maturation of the Toba magma reservoir

References:

Caricchi, L., Simpson, G. and Schaltegger, U., 2014. Zircons reveal magma fluxes in the Earth's crust. Nature, 511(7510), pp.457-461.

Caricchi, L., Simpson, G. and Schaltegger, U., 2016. Estimates of volume and magma input in crustal magmatic systems from zircon geochronology: the effect of modeling assumptions and system variables. Frontiers in Earth Science, 4, p.48.

Keller, C.B., Schoene, B. and Samperton, K.M., 2018. A stochastic sampling approach to zircon eruption age interpretation. Geochemical Perspectives Letters (Online), 8(LLNL-JRNL-738859).

Miller, J.S., Matzel, J.E., Miller, C.F., Burgess, S.D. and Miller, R.B., 2007. Zircon growth and recycling during the assembly of large, composite arc plutons. Journal of Volcanology and Geothermal Research, 167(1-4), pp.282-299.

Schmitt, A.K., Sliwinski, J., Caricchi, L., Bachmann, O., Riel, N., Kaus, B.J.P., Cisneros de Léon, A., Cornet, J., Friedrichs, B., Lovera, O. and Sheldrake, T., 2023. Zircon age spectra to quantify magma evolution. Geosphere, 19(4), pp.1006-1031.

Samperton, K.M., Schoene, B., Cottle, J.M., Keller, C.B., Crowley, J.L. and Schmitz, M.D., 2015. Magma emplacement, differentiation and cooling in the middle crust: Integrated zircon geochronological–geochemical constraints from the Bergell Intrusion, Central Alps. Chemical Geology, 417, pp.322-340.

Tavazzani, L., Wotzlaw, J.F., Economos, R., Sinigoi, S., Demarchi, G., Szymanowski, D., Laurent, O., Bachmann, O. and Chelle-Michou, C., 2023. High-precision zircon age spectra record the dynamics and evolution of large open-system silicic magma reservoirs. Earth and Planetary Science Letters, 623, p.118432.

Watson, E.B., 1996. Dissolution, growth and survival of zircons during crustal fusion: kinetic principals, geological models and implications for isotopic inheritance. Earth and Environmental Science Transactions of the Royal Society of Edinburgh, 87(1-2), pp.43-56.

Weber, G., Caricchi, L., Arce, J.L. and Schmitt, A.K., 2020. Determining the current size and state of subvolcanic magma reservoirs. Nature Communications, 11(1), p.5477.

**Authors' Response to Reviewer 2 of**

**A statistical analysis of zircon age distributions in volcanic, porphyry and plutonic rocks**

Chetan Nathwani, Dawid Szymanowski, Lorenzo Tavazzani, Sava Markovic, Adrianna L. Virmond, and Cyril Chelle-Michou
*Geochronology Discussions,* `https://doi.org/10.5194/gchron-2024-25`
* * *
**RC**: Reviewers' Comment,    **AR**: Authors' Response

We thank the reviewer for their supportive and constructive feedback on our manuscript and we are pleased they find it to be useful contribution to the geochronology community. They provide a series of very useful minor comments on the manuscript which we respond to individually below.

General Comments:

This manuscript addresses a critical issue in modern igneous petrology: How best to understand the measured timescales of crystallization in igneous rocks. The lens here is an important subset of this problem, zircon crystallization timescales, when measured using whole grain analyses at precisions that are much higher than the crystallization timescales.

In this work, the authors produce a large number of carefully curated U-Pb datasets, present a quantitative model for comparing their distributions (Wasserstein difference) and reducing the degrees of freedom in them (PCA), extract trends in the data that correlate broadly with the type of igneous system, and then compare these trends with predictions they make based on simple forward models. I found it to be a very clearly written paper, with good explanations of the quantitative techniques and explanations of how they related to the underlying datasets. The figures are high quality, the writing is excellent, and the choice of topic is well within the remit of Geochronology.

Specific Comments:

Like I said above, this is a very good paper. The only real issue is that the authors don't address that there are important differences between their forward models and real processes that drive crystallization and preservation of crystallization age distributions. As I'm writing this I note that this is something that the first reviewer also pointed out, so I won't spend a lot of time on it, but I think this is probably worth addressing qualitatively, for completeness. Both in terms of what the T-t history is like, and additional complications such as those highlighted by Klein and Eddy, and biases in the measured zircon record. I appreciate that it may not be possible to include those complications at this time in a quantitative model, but they are both real and important.

We have incorporated more of a discussion on the differences between our modelling and real processes, including much more detail on what could explain the difference between the results of our statistical analysis and thermal models from other studies. A more detailed response can be found in our reply to Reviewer 1. We have also made the reader aware of the caveat of our study that it does not consider some of the complications highlighted by Klein and Eddy (see below).

Specific comments/technical corrections

L43-44: It's worth noting that sampling biases may also play a role. I can't speak for every analyst, of course,

but typically the largest and highest quality crystals are selected from a population. The size bias alone may be significant, as is the fact that whole grains modified by CA bias U-Pb dates by U concentration, dissolve components that are accessible to HF during the partial dissolution phase of CA, and integrate the remaining grain.

We agree that it would be useful to make a note of some additional biases. We have thought not to get into too much detail about human biases invoked by ID-TIMS as this is something that will not be unique to a specific type of sample, or the chemical abrasion biases as we do not want to convolute and confuse the paper. We do think however that it is worth noting the fact that there is a bias of bulk grain analyses to younger ages due to the volumetric differences in core to rim, as pointed out by Klein and Eddy (2023. We specify therefore that our model assumes instantaneous crystallisation of zircon

L120: This is an excellent summary of the method.

Thank you.

L134-137: I think it's fair to stick to one type of dataset but the objections to 230/238 geochronology aren't strong. Variable and asymmetric uncertainties aren't difficult to deal with, and initial [230/238] are determined in many high quality datasets (and doesn't vary infinitely). There are good reasons that plutonic datasets are rare (or absent?) and that's because the technique is exclusively applicable to young rocks, and that's a reason enough alone to exclude it.

This is fair. By far the most important reason to not include these data is their being limited to volcanic samples (simply because there are few if any plutons this young and already exposed). The asymmetric uncertainties are not a big deal, but the lack of a robust way to estimate melt-(230/238) corresponding to each zircon date remains a major source of uncertainty. We have adjusted this part to reflect this in the revised manuscript.

L159: This is very good.

Thanks!

L201-206: I'm a bit confused by this paragraph and unsure as to what I'm supposed to take away from it as a reader. It describes a technique that is not used, justifies why it is suboptimal, but then in the last sentence tells me that the results are the same? I have no specific objection here but I think the authors might want to clarify for a reader what they should take away from this. For me personally, as a geochemist who has used MDS in the past, I'd be just as happy if this paragraph didn't exist? I don't think many readers will look at this and wonder why they didn't use MDS.

This is a fair point that it does not necessarily add anything to the manuscript and the reader will not wonder why the readers did not use MDS. We therefore have accepted the reviewer's suggestion and removed this from the revised manuscript.

L220-221: Minor point but mixing and matching confidence/coverage intervals is confusing to a reader (the parameter fit uncertainties are at a coverage factor of 1 and the result is at 2).

Yes, this is a valid point thanks, the standard errors on the fit parameters have now been updated to 2SE in line with the $2\sigma$ of the output.

L221: This is a really useful result, and I could see it being something used by others in future work – it might be worth adding a bit more context for those who do? Can you specify the time range that this is applicable to, and if the fit parameters uncertainties are correlated? In the introduction you list 0.1-4567 Ma, but that's

probably not relevant here (if I do the calculation right it's 40% at 0.1 Ma.)

This is calculated over the range of dates included in the compilation we used (Markovic et al.) but only up to 1000 Ma (to keep it relevant to the maximum important for our study and to avoid the fact that older dates would use Pb/Pb). We have specified this now in the revised manuscript. We agree it can be useful if one is interested in the general development of uncertainties with age, but we would not take it too far. Specifically, this shows the mid-range uncertainties in dates from multiple labs, and it uses a relatively simple fit. So real uncertainties available to any given lab may be much better or much worse.

It's also worth noting that these uncertainties differ markedly from those listed rather optimistically on line 138 (e.g., if I did the calculation correctly, it bottoms out at 0.11% in the Neoproterozoic). I'd recommend revisiting line 138 to bring it in line with the actual uncertainties parameterized in this equation – that's more of a boilerplate ID-TIMS boast than a realistic assessment of actual typical precisions from young zircon populations, where for even low-blank labs, the precision ends up being limited by the blank isotopic composition variability.

While the value in line 138 is very much "a boilerplate ID-TIMS boast", a realistic assessment of typical precisions based on that same compilation shows that it's actually fairly close to truth for all but the very youngest samples - see Fig. 11e in Markovic et al. We will adjust line 138 to reflect that reality better.

L235-240: It's worth revisiting the issues raised by Klein and Eddy at this point. Not that they need to be incorporated but just as a caveat for the reader.

Yes agreed that it would be useful to mention their work. We have decided it would best to bring this into the Methods, where we state that our model, for simplicity, assumes the instantaneous growth of each zircon and ignores the protracted growth of each crystal and the inherent bias of bulk grain analysis to younger ages. We also add a reference to Melnik and Bindeman (2018) which discusses this.

L268: Please be cautious here in assuming that natural processes won't result in gaussian distributions. The central limit theorem applies to natural processes as well – the samples we recover average a range of disparate pieces of data and proceses, and averaged samples from any distribution are gaussian.

Very fair point. This requires some careful rewording. What we mean here is that our threshold for $\Delta T/\sigma$ is probably valid as the lowest $\Delta T/\sigma$ in our filtered datasets still exhibit skew indicating they are not dominated by gaussian uncertainty (and would thus typically show normal distributions). We have rephrased to remove the statement of non-normal distributions being dominated by geological dispersion.

L305: Can you compare this to other estimates if you are aware of any? (This isn't a leading question, I'm can't find one myself)

There are some estimates made by Caricchi et al. (2016) who compared the mean, mode and median of a zircon age distribution with different numbers of zircon sampled. When sampling a normal distribution, they show that already with 10 zircons sampled, the mean, mode and median are similar to the underlying distribution. For a skewed distribution (which is what our sampling is performed on), they show that significant improvement is made when up to 30 zircons are sampled. Their sampling is performed at 100 kyr and we are unclear how/if uncertainties are propagated onto these results. Our study of the effect of number of zircons sampled is based on comparing the entire shapes of distributions rather than mean, mode and median. Given the difference in approaches and different ages under which the modelling is performed, we decided not to make a comparison in the text here as getting into the details will distract a bit the reader from the key point.

L412: This seems like an unnecessary swipe at LA-ICPMS labs. Using data that is accurate (including

accurate uncertainties) is obviously critical and there's no reason to cast aspersions on a specific technique. Peak hopping TIMS Pb data on Phoenix instruments have uncertainties on isotope ratios underestimated by 1.7x due to autocorrelation (and some other factor on any other TIMS), but that fact is equally out of place here. LA-ICPMS is an easy target because of the large number of practitioners and therefore the larger n of outliers.

We do not believe we are making an unwarranted swipe at the LA-ICP-MS technique but we think it is very important to note here that using spot analyses to reconstruct an age distribution in these young rocks may be more challenging than one might think from just looking at the reported uncertainties. Quite a lot of studies report single-spot $2\sigma$ uncertainties at about 0.5% (which is likely just considering analytical uncertainty), but when taking into account repeatability of reference material analyses (which is a good approximation for systematic uncertainties asscociated with 'matrix effects' in LA-ICP-MS analyses) this is actually within the range of 2-4% which is quite a substantial difference (Sliwinski et al., 2022). Hence, we feel it appropriate here to mention as we want to avoid people overinterpreting age distributions from LA-ICP-MS data with under-estimated uncertainties. We have replaced the word "neglected" with "do not account for" in order to soften these words for the LA-ICP-MS community.

The issue of potentially underestimated uncertainties in peak-hopping TIMS Pb analyses is not insignificant as it will affect our threshold $\Delta t/\sigma$ by reducing the resolution of some datasets. It is however hard to address for now since it is not documented to a point where we could, for example, apply a correction factor to each dataset. We hope progress in data reduction algorithms for ion counter analyses will soon catch up with the needs of the community in that regard.